# Automated Benchmark Generation for Repository-Level Coding Tasks

**Konstantinos Vergopoulos** [1] [*]  **Mark Niklas Müller** [1] [*]  **Martin Vechev** [1] [2]

## Abstract

Code Agent development is an extremely active research area, where a reliable performance metric is critical for tracking progress and guiding new developments. This demand is underscored by the meteoric rise in popularity of SWE-Bench– a benchmark that challenges code agents to generate patches addressing GitHub issues given the full repository as context. The correctness of generated patches is then evaluated by executing a human-written test suite extracted from the repository after the issue's resolution.

However, constructing benchmarks like SWE-Bench requires substantial manual effort to set up historically accurate execution environments for testing. Crucially, this severely limits the number of considered repositories, e.g., just 12 for SWE-Bench. Considering so few repositories, selected for their popularity runs the risk of leading to a distributional mismatch, i.e., the measured performance may not be representative of real-world scenarios running the riks of misguiding development efforts.

In this work, we address this challenge and introduce SETUPAGENT, a fully automated system capable of historically accurate dependency setup, test execution, and result parsing. Using SETUPAGENT, we generate two new datasets: (i) SWEE-Bench an extended version of SWE-Bench encompassing hundreds of repositories, and (ii) SWA-Bench a benchmark focusing on applications rather than libraries. Comparing these datasets to SWE-Bench with respect to their characteristics and code agent performance, we find significant distributional differences, including lower issue description quality and detail level, higher fix complexity, and most importantly up to 60% lower agent success rates.

---
[*]Equal contribution [1]LogicStar AI [2]Department of Computer Science, ETH Zurich. Correspondence to: Mark Niklas Müller <mark@logicstar.ai>.

*Proceedings of the $42^{nd}$ International Conference on Machine Learning*, Vancouver, Canada. PMLR 267, 2025. Copyright 2025 by the author(s).

## 1. Introduction

Code Agents are quickly becoming one of the most promising and actively researched applications of Large Language Models (LLMs); partly due to their potential to revolutionize the 700 billion dollar software industry (Statista). To measure progress and more importantly steer further developments in this field, high-quality datasets and benchmarks are crucial. In particular, it is essential that they are representative of real-world use cases, sufficiently large to allow meaningful statistical analysis, and diverse and recent enough to avoid unintentional overfitting and contamination.

**Existing Benchmarks** However, function-level benchmarks like HumanEval (Chen et al., 2021), popular for evaluating LLM's coding performance, are unrepresentative of real-world use, lack diversity, and are becoming saturated. To address these limitations, SWE-Bench (Jimenez et al., 2024) was proposed as the first repository-level coding benchmark based on real-world tasks, i.e., resolving GitHub issues. Yet, it still suffers from several limitations. (i) It is limited to few repositories, potentially leading to overfitting to these specific codebases. (ii) Its sole focus on libraries in contrast to applications raises generalizability questions. (iii) Its focus on popular repositories not only makes it less representative but also increases the chances of contamination with general codebase knowledge. (iv) Its static nature leads to most or even all instances being created before recent models' knowledge cutoff, allowing even the exact instances to be present in the training data.

**Creating Repository-Level Benchmarks** To address these challenges, we would like to create more diverse benchmarks and update them frequently with new tasks. However, while the GitHub Issues and Pull Requests (PRs), serving as task descriptions and reference solutions, respectively, for SWE-Bench-like benchmarks can be scraped automatically, evaluating the correctness of a solution, requires the repository's test suite to be executed. This, in turn, requires setting up historically accurate execution environments, identifying the correct test commands, and parsing the results. Prior work addressed this problem either manually (Jimenez et al., 2024) or by aggressively filtering out instances where default commands were unsuccessful (Jain et al., 2024c). However, both approaches yield limited diversity and don't lend themselves to frequent updates.

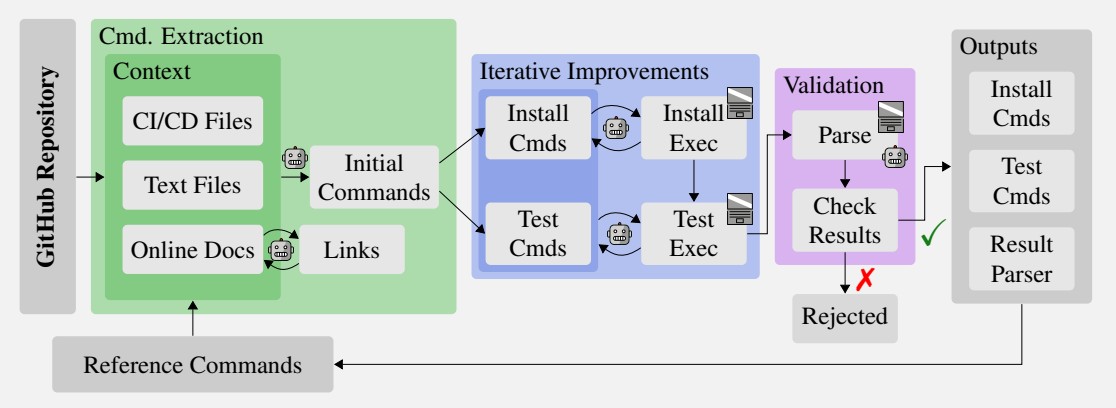

*Figure 1.* Overview of SETUPAGENT where a 🤖-icon represents an LLM driven step and a 🖥-icon represents execution feedback.

**This Work: SETUPAGENT** To address this challenge, we propose SETUPAGENT, the first method to automate this setup process, enabling us to create repository-level code benchmarks fully automatically from a list of GitHub repositories. SETUPAGENT works in three key phases (illustrated in Figure 1): (i) Command Extraction (green ■ in Figure 1), (ii) Iterative Testing and Improvement (blue ■), and (iii) Validation (purple ■). In the extraction phase, SETUPAGENT analyzes relevant context, such as README.md files, CI/CD configurations, and referenced web pages, to propose installation and testing commands. During the iterative improvement phase, SETUPAGENT then executes these commands in a clean environment and leverages an LLM to systematically diagnose and resolve issues. Finally, in the validation phase, SETUPAGENT ensures that the generated commands are reliable by verifying the correctness of the setup based on test results, only accepting configurations that meet a predefined success threshold.

**This Work: Generated Benchmarks** We demonstrate SETUPAGENT's capability to generate coding benchmarks from a list of repositories by creating SWA- and SWEE-Bench, each addressing specific shortcomings of SWE-Bench. Both are designed to be representative of real-world use cases, consider many repositories leading to diverse benchmarks, and can be frequently updated without manual effort to avoid contamination and overfitting. SWA-Bench focuses on software applications, containing 44 projects while SWEE-Bench focuses on diversity and less popular projects containing 366 Python repositories. Comparing SWA- and SWEE-Bench to SWE-Bench, we find significant distributional differences, including lower repository age and popularity at issue creation, a larger focus on recent issues, and significantly more complex reference code fixes (2-4x more modified files and lines). Evaluating popular code agents on these datasets, we find significant performance differences for some models and statistically significant signs of contamination, highlighting the importance of evaluating on representative benchmarks.

**Key Contributions** of this work are:

- We propose SETUPAGENT, the first method for autonomously creating historically accurate execution environments.

- We leverage SETUPAGENT to create two datasets for repository-level code generation SWA- and SWEE-Bench, focusing on applications and diverse projects, respectively.

- We extensively analyze SWA- and SWEE-Bench in terms of their characteristics and corresponding code agent performance.

## 2. Related Work

**Code Agents** To fully leverage the potential of LLMs for code generation, they have been equipped with tools to interact with their environment without additional user input, e.g., by searching, viewing, and editing code, (Wang et al., 2024a). These so-called code agents have shown great promise on complex tasks (Bouzenia et al., 2024a; OpenDevin, 2024; Zhang et al., 2024; Yang et al., 2024b; Xia et al., 2024; Aider, 2024; Ridnik et al., 2024; Wang et al., 2024b). In this work, we evaluate some of the best-performing open-source agents.

**Code Generation Benchmarks** With the success of LLMs in the domain of code generation, an increasing variety of function-level code generation benchmarks were proposed to assess their capabilities (Chen et al., 2021; Hendrycks et al., 2021; Austin et al., 2021; Jain et al., 2024a; Huang et al., 2024). However, not only were these increasingly saturated by state-of-the-art models but their focus on interview-style function-level coding challenges makes them also unrepresentative of the complexities of real-world codebases and software engineering tasks.

To address these limitations, a range of repository-level code-generation benchmarks have been proposed recently (Liu et al., 2023; Jain et al., 2024b; Jimenez et al., 2024).

However, a repository-level context not only makes code generation but also dataset generation more challenging as it requires a historically accurate execution environment to be set up, the project's test suite to be run, and detailed results to be extracted. The required manual effort led to existing datasets focusing on a relatively small number of popular repositories. As a result, they are prone to overfitting, often lack diversity, and can easily contaminate the training data.

**Automatic Dataset Generation** These challenges could be addressed via automatic dataset generation, which has been successfully applied to function-level benchmarks by scraping tasks from coding challenge websites and doing varying levels of manual post-processing (Hendrycks et al., 2021; Jain et al., 2024a; Huang et al., 2024).

Jimenez et al. (2024) transfer these ideas to repository-level benchmarks, automatically scraping GitHub repositories, issues, and pull requests resolving these issues to create SWE-Bench consisting of 12 repositories and 2294 instances. However, they still created the required execution environments and test commands manually. Further, the resulting issues were shown to suffer from underspecified descriptions and overly specific tests (Chowdhury et al., 2024).

Jain et al. (2024b) create R2E, a function-level synthesis benchmark with repository context by scraping GitHub repositories and masking out the function to be generated. They automated the setup by applying a default approach for projects with a `setup.py` or `pyproject.toml` file, automatically generating equivalence tests, and filtering out all instances where this approach fails. However, this approach aggressively filters projects with more complex installation procedures, not only introducing a selection bias but also yielding only 246 instances.

In this work, we combine the more interesting repository-level tasks with a fully automated benchmark generation process, by introducing and leveraging SETUPAGENT to automatically extract the installation and testing procedures for every task instance, allowing us to create larger and more diverse benchmarks efficiently.

Bouzenia & Pradel (2024), concurrently proposed EXECUTIONAGENT, a tool to automatically set up and test repositories. However, it is 60 times slower than SETUPAGENT, does not support historical states, and does not extract results at test-level granularity. Even if the latter two shortcomings were addressed, it would remain infeasibly slow taking, e.g., over 4 months to generate SWEE-Bench[1].

# 3. Autonomous Environment Setup

In this Section, we first outline the requirements for a setup and testing agent to be used for benchmark generation and then describe the agent we develop for this purpose.

---

[1]Extrapolated from ∼150 repositories.

## 3.1. Notation and Definitions

We first introduce notation to describe repository-level coding tasks, adapted from Mündler et al. (2024). Given a codebase $R$, we obtain $R \circ X$ by applying the code patch $X$. We similarly denote the test suite $T$ with $T \circ S$ after applying the test patch $S$. A single test $t \in T$ can either pass (P) or fail (F) when executed against the codebase $R$ in an execution environment $E$. We write: $\text{exec}_E(t, R) \in \{P, F\}$ and let the order $P > F$ hold.

A repository-level coding task can be written as the tuple $(R, T, I, E, S^*, X^*)$, where $R$ and $T$ are the original codebase and test suite, respectively, $I$ is the issue description, $E$ the execution environment, and $S^*$ and $X^*$ the reference test and code patch, respectively. By executing all tests $t_i \in T \circ S^*$ in the execution environment $E$, first against the original ($R$) and then the patched codebase ($R \circ X^*$), we obtain the reference test behavior $b_i^* = (\text{exec}_E(t_i, R) \rightarrow \text{exec}_E(t_i, R \circ X^*))$. We call $t_i$ with, e.g., $b_i^* = F \rightarrow P$ a fail-to-pass test as it fails before the reference fix is applied but passes afterward. We let the partial order $F \rightarrow P > F \rightarrow F$ and $P \rightarrow P > P \rightarrow F$ hold.

The task is now to generate a patch $X'$, given only $(R, T, I, E)$, such that the test behavior $b_i' = \text{exec}_E(t_i, R) \rightarrow \text{exec}_E(t_i, R \circ X')$ matches or improves on the reference result, i.e., $b_i' \geq b_i^*$ for all tests $t_i \in T \circ S^*$.

## 3.2. Setup Agent Requirements

A generic setup agent targeting individual, up-to-date repositories only has to satisfy one main requirement: *Correctness* – It must extract and run the installation and testing commands before parsing the test results. However, benchmark generation, i.e., generating the execution environment $E$ given the remaining components of a coding task, imposes additional requirements: *Historical Accuracy* – Benchmark instances are based on specific, often outdated versions of a codebase $R$. The execution environment $E$ must thus use historically accurate dependency versions to reproduce the original issue faithfully and avoid version incompatibilities. *Efficiency* – To generate a dataset of many hundreds of instances, the setup agent must be efficient enough to keep total runtime reasonable (hours or at most few days). *Granularity* – Evaluating agent success requires test-level results to be parsed from the test suite output.

## 3.3. SETUPAGENT

**Overview** SETUPAGENT works in three phases illustrated in Figure 1: (1) Extraction (■ in Figure 1), (2) Iterative testing and imporvement (■), and (3) Validation (■). In the first phase, SETUPAGENT extracts a first version of the installation and testing commands from all relevant files, referenced webpages, and, if available, successful commands

from similar versions of this repository. In the second phase, SETUPAGENT iteratively executes first the installation and then testing commands, analyses the results and updates the commands. Finally, in the third phase, SETUPAGENT validates the resulting commands by executing them, extracting the test results, and rejecting the proposed commands, if too few tests pass. Validated commands are then returned to the user and saved in a reference database to facilitate installations of different versions of the same repository.

```
Input:
Please extract all commands required to install
<project_name> in a clean environment and run its
test suite from the context below.
'''
## README.md
<file_content>
## CONTRIBUTING.md
<file_content>
'''

LLM Response:
'''bash
apt-get install -y graphviz    # installation
pip install -r req.txt         # installation
nox -e test                    # testing
'''
Installation Commands:
apt-get install -y graphviz
uv pip install -r req.txt -exclude-newer 2022-04-11
Testing Commands:
nox -e test
```

*Figure 2.* Illustration of the extraction phase of SETUPAGENT. Please see App. B for the full-length prompts.

**Extraction Phase** The goal of the Extraction Phase is to extract an initial version of the installation and testing commands. We illustrate it in Figure 2, using a significantly simplified prompt and shortened response for illustration purposes. In the first step, SETUPAGENT reviews all relevant files to identify the correct Python version. If available, it is directly parsed from the `setup.py` or `pyproject.toml` file, otherwise version constraints are extracted from different sources using an LLM and then resolved while taking issue creation time into account to ensure *Historical Accuracy*. Next, SETUPAGENT uses a heuristic to identify possibly relevant context like `README.md` and `CONTRIBUTING.md` files, CI/CD configurations, and referenced webpages. It then reviews these sources and directly rejects those not containing information relevant to installation or testing, thus keeping its context size small and improving *Efficiency*. Finally, SETUPAGENT prompts an LLM to extract the installation and testing commands from the chosen sources. To ensure *Historical Accuracy*, SETUPAGENT enforces the use of the `uv` environment manager for Python dependencies, configuring it to exclude dependency versions released after the issue creation.

**Iterative Improvement Phase** SETUPAGENT now tries to install and test the project in a clean Docker container using the extracted commands. If any of the installation steps fail, i.e., exit with a non-zero exit code, or the test

```
Input:
You are trying to install and test <project_name>
using the following commands.
'''bash
<commands>
'''
Modify the commands to resolve the following error:
'''
...
ModuleNotFoundError:  No module named 'rustworkx'
'''

LLM Response:
We have to install rustworks first:
'''bash
pip install rustworkx
'''
Updated Installation Commands:
apt-get install -y graphviz
uv pip install -r req.txt exclude-newer 2022-04-11
uv pip install rustworkx exclude-newer 2022-04-11
```

*Figure 3.* Illustration of the iterative improvement phase of SETUPAGENT, where the error message was obtained by executing the commands from the previous iteration.

commands fail with an error, SETUPAGENT selects the relevant sections of the error messages and then prompts an LLM to identify the issue and propose a modification of the installation commands to fix it. We illustrate this in Figure 3, where SETUPAGENT adds a missing dependency. Now SETUPAGENT tries to resolve issues apparent in the test results, analyzing the logs to determine whether a failure is due to incorrect test commands, an incorrect installation, or a bug in the codebase. Depending on the result, SETUPAGENT prompts an LLM to modify the installation or testing commands to fix the issue or passes the repo on to the validation phase. This iterative improvement is repeated until all errors are resolved or an iteration limit is reached. Using a moderate iteration limit of 4 steps, we achieve significantly improved *Correctness* without sacrificing *Efficiency*.

```
Input:
Please assess whether <project_name> was installed
and its test suite executed correctly given the
resulting printout.
Answer YES or NO.
'''
...
===== 2597 passed, 3 failed in 10.85s =====
'''

LLM Response:
YES
```

*Figure 4.* Illustration of the first step in the Validation phase.

**Validation Phase** In the validation phase, SETUPAGENT first queries an LLM to assess whether the installation and testing were successful, illustrated in Figure 4. If the LLM judges the installation to be successful, SETUPAGENT updates the test framework's configuration to return test-level results, e.g., by adding `-rA` to a `pytest` command, thus ensuring *Granularity*. It then selects the correct parser from a pre-defined set to extract test-level results and checks the number of passing and failing tests. We consider the installation to be successful if at least 95% of tests pass.

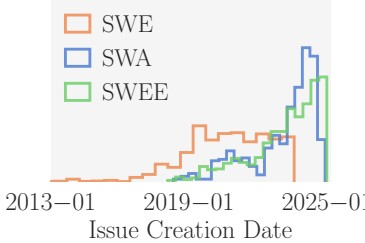 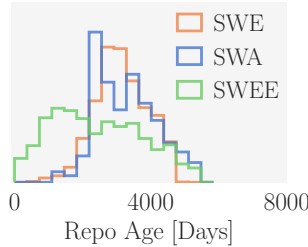 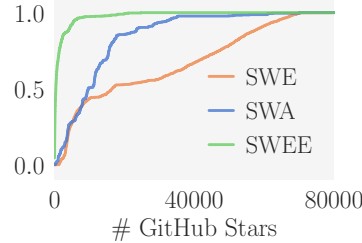

*Figure 5.* PDFs (left and middle) and CDF (right) of PR creation dates (left), repository age at PR creation time (middle), and number of GitHub stars (right) for SWA, SWEE, and SWE-Bench.

## 4. Code Generation Benchmarks

In this Section, we describe how we leverage SETUPAGENT to create SWA- and SWEE-Bench, two new benchmarks addressing specific limitations of SWE-Bench. We compare these datasets with SWE-Bench and provide insights into distributional differences.

**Automatically Generated Benchmarks** By creating execution environments automatically, we address two core limitations of manually generated repository-level benchmarks: (i) we can consider many more repositories without requiring infeasible manual labor, thus improving diversity and reducing the risk of overfitting and (ii) we can easily update benchmarks by creating new tasks from recent PRs and issues, thus ensuring that models are not contaminated with benchmark instances (see Figures 5 and 6).

**SWA-Bench** Many practitioners using Code Agents develop software applications that suffer from different types of bugs compared to libraries due to architectural and structural differences. As SWE-Bench only considers libraries, we design SWA-Bench to focus only on applications.

**SWEE-Bench** We observe that more popular repositories tend to have higher-quality codebases and issue descriptions. This includes, e.g., a more consistent (file) structure and naming conventions, better documentation including detailed docstrings for most functions, and issue descriptions following a precise template (see Figures 7 and 8). As SWE-Bench focuses on particularly popular Python repositories, the resulting tasks can be unrepresentative of real-world use. Therefore, we design SWEE-Bench with a focus on diverse and less popular (median of 365 vs 16k stars) Python repositories (see Figure 5).

### 4.1. Dataset Creation

**Source Repositories** For SWA-Bench, we combine a list of 468 popular Python applications (Hashemi, 2024) with a list of 50 Python projects from Bouzenia et al. (2024b), leading to a total of 475 candidate repositories after deduplication. For SWEE-Bench, we consider the 8000 most

*Table 1.* SWEE pipeline from projects to tasks. A PR is valid if it resolves an issue, modifies a test file, and is merged. An instance valid, if it has additionally at least one $F \rightarrow P$ test.

| Step | # Repos | # PRs |
|---|---|---|
| Initial Projects | 8000 | |
| + GH Repo Found | 7057 | |
| + Preprocessing | 5097 | |
| + Permissive License | 3800 | |
| + Has valid PR | 2377 | |
| + SETUPAGENT succeeds | 514 | |
| + Get $n_{per\_repo}$ valid PRs | | 2115 |
| + SETUPAGENT succeeds | | 1513 |
| + valid instance | | 885 |

downloaded PyPi projects at the time (van Kemenade et al., 2024) with between 100k and 1.5B monthly downloads and 0 to 25k stars, leading to good diversity while focusing on relevant projects.

**Dataset Creation with SETUPAGENT** We combine the original PR filtering process from Jimenez et al. (2024) with our SETUPAGENT as follows: For every project, we first locate the corresponding repository, deduplicate the results, and filter out repositories that are not published under a permissive license. We then scrape issues and pull requests for each repository until we find the most recent PR that is merged, resolved an issue, and modified a test file. We call this a valid PR. We then use SETUPAGENT to set up an execution environment $E$ for the corresponding codebase $R$ (see Section 3). For repositories where this succeeds, we scrape additional PRs until we have $n_{per\_repo}$ valid ones or, for SWEE, reach a maximum of 500 PRs. We then use SETUPAGENT to create the execution environment $E$ for each corresponding codebase $R$ in reverse chronological order per repository, populating SETUPAGENT's reference commands database to speed up the setup process. Finally, we split every PR into a reference code patch $X^*$ and test patch $S^*$. We execute the full test suite $T \circ S^*$ before and after the code patch is applied, i.e., on $R$ and $R \circ X^*$, respectively, to obtain the reference test behaviors $b_i^*$. We then filter out PRs, where test execution fails in one of these settings or which have no $F \rightarrow P$ test, i.e. $\nexists t \in$

$T \circ S^* : \exec_E(t, R) \to \exec_E(t, R \circ X^*) = F \to P$. The remaining PRs form the valid instances of the generated benchmark. We choose $n_{per\_repo} = 50$ for SWA-Bench and $n_{per\_repo} = 10$ for SWEE-Bench to obtain the desired number of tasks and show the number of repositories and PRs this leads to in Tables 1 and 10, respectively.

**Ease of Use** To make benchmark generation and use as easy as possible, SETUPAGENT only requires a list of repositories to generate a dataset in a format compatible with SWE-Bench along with docker images with all dependencies installed. We publish SWA-Bench on HuggingFace and the corresponding docker containers at logicstarai/swa-bench. A suitable evaluation harness is available at github.com/logic-star-ai/SWEBench.

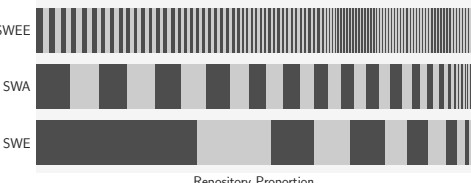

*Figure 6.* Comparison of the repository distribtuion of SWEE-, SWA-, and SWE-Bench across instances.

## 4.2. Benchmark Characteristics

**Diversity** We compare the distribution of instances over repositories in Figure 6 and observe that while instances in SWE are heavily concentrated in only a few repositories, with over 50% of instances belonging to only two out of 12 total repositories, SWA- and SWEE-Bench show much more diversity with 535 instances from 44 repositories and 885 from 366 repositories. See App. C for a full list.

**Codebase Characteristics** We compare benchmarks with respect to codebase characteristics in Table 2 and Figure 5 and observe that SWE-Bench, compared to, SWA- and especially SWEE-Bench contains significantly older and more popular (# GitHub stars) repositories and larger, more complex codebases (# files and # lines of code).

**Issue Description Quality** To assess the issue description quality, we measure the number of words, error messages, and code blocks they contain as well as the overlap between the files mentioned there and modified in the reference fix and the overlap between the issue description and the reference solution itself. We show cumulative distribution functions (CDFs) of the aforementioned characteristics in Figure 7. We observe that while SWA-Bench has more detailed issue descriptions (longer, more code blocks, and more error messages), they do not seem to be of higher quality (less overlap with the reference solution and equal file mentions). Comparing SWE-Bench and SWEE-Bench, we

*Table 2.* Comparison of mean dataset characteristics.

|  |  | SWA | SWEE | SWE |
|---|---|---|---|---|
| Codebase | # Files | 899 | 77 | 1491 |
|  | # Lines | 112k | 14.8k | 321k |
| Issue Descriptions | # Words | 240.2 | 125.1 | 181.3 |
|  | # Error Messages | 0.20 | 0.13 | 0.19 |
|  | # Code Blocks | 1.53 | 1.19 | 1.06 |
| Tests | # $P \to P$ | 564.2 | 226.6 | 120.1 |
|  | # $F \to P$ | 38.8 | 38.1 | 13.5 |
|  | # $F \to F$ | 3.7 | 1.4 | 3.4 |
|  | # $P \to F$ | 0.11 | 0.03 | 0.04 |
| Test Patches | # Edited Files | 1.89 | 2.05 | 1.52 |
|  | # Edited Lines | 74.8 | 91.5 | 39.2 |
|  | # Added Tests | 9.10 | 23.78 | 6.37 |
|  | # Removed Tests | 16.77 | 2.49 | 0.54 |
| Fix Patches | # Edited Files | 3.26 | 3.26 | 1.66 |
|  | # Edited Lines | 104.3 | 169.9 | 41.0 |

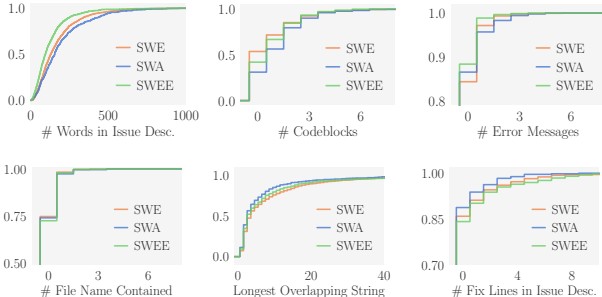

*Figure 7.* CDFs over issue description characteristics. Number of words (top left), number of code blocks (top middle), number of error messages (top right), number of filenames contained in the issue description and modified in the reference solution (bottom left), the overlap between the issue description and the reference solution in terms of longest string match (bottom middle) and complete lines (bottom right). A CDF further down and to the right indicates a higher value.

observe longer issue descriptions and slightly more overlap with the reference solution in SWE-Bench but otherwise similar characteristics.

**Fix Complexity** To assess the complexity of required fixes, we measure the number of lines and files modified in the reference solution and the number of tests that flip from passing to failing (and vice versa). We show CDFs in Figure 8 and observe that while SWEE- and SWA-Bench have similar distributions across all these metrics, SWE-Bench fixes are significantly less complex by all metrics.

## 4.3. Manual Review

While SETUPAGENT ensures that at least 95% of tests pass, some quality issues may still remain. To assess their frequency and impact, we conduct a manual review focusing

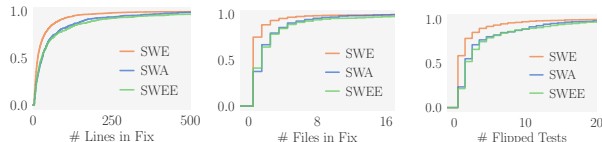

*Figure 8.* CDFs over fix-complexity characteristics. Number of edited lines (left), number of edited files (top middle), number affected tests, i.e., $F \rightarrow P + P \rightarrow F$ (right). A CDF further down and to the right indicates higher characteristic values.

on two types of issues: (i) dataset quality along the lines of SWE-Bench Verified (Chowdhury et al., 2024), i.e., whether the issue is sufficiently well-specified to be resolved without further information and whether the tests are suitable to check its resolution and (ii) setup success, i.e., whether SETUPAGENT was able to set up the environment correctly and run the tests as expected.

**Dataset Quality** In line with prior work (Chowdhury et al., 2024), we focus on two main aspects: (i) task specificity, i.e., whether the issue is sufficiently well-specified to resolve it without further information and (ii) test quality, i.e., whether the added unit tests check for the described behaviour or are overly specific to concrete implementations such as exactly matching error messages. We assess both of these criteria on a scale from 0 to 3, where 0 is the best score and 3 the worst, using the same annotation guide as Chowdhury et al. (2024). That is, we scored issue specificity and clarity on a scale of 0 (well-specified issue with clear success criteria) to 3 (almost impossible to solve correctly without further instructions) and test quality from 0 (test perfectly covers valid solutions) to 3 (tests are too narrow or broad or requiring information not provided in the issue description). On both scales, 0 and 1 are considered acceptable, while 2 and 3 are insufficient.

We conducted a manual review of 30 randomly chosen SWA instances and observed the following: 21 (70%) instances have a meaningful and sufficiently complete issue description, and 20 (67%) of these additionally have suitable tests (27 or 90% across all instances) to check whether the issue was fixed. This is in line with the results of Chowdhury et al. (2024) and shows that the majority of issues are solveable, with current performance levels still leaving significant room for improvement, making SWA a suitable benchmark for current and future code generation systems.

**Setup Success** To validate the automated assessment used by SETUPAGENT to determine whether an instance was setup correctly, we assess both the extracted setup and testing steps as follows. (i) We score the setup on a binary scale of 0 (correct setup that is functionally equivalent with the described setup and 1 (incorrect setup). (ii) We score testing on a scale from 0 (functionally equivalent to described testing) to 2 (tests only partially or not at all executed), where

both 0 and 1 are considered acceptable.

We manually review the same 30 instances as above and observe the following: All instances run the correct tests with 23 (77%) using exactly the same test commands as provided in the reference. 22 (73%) of these instances additionally have a fully correct installation/setup. These results indicate that SETUPAGENT has no trouble extracting the correct testing steps, with the setup proving slightly more challenging but also forgiving considering that 95% of tests passed despite minor setup errors.

## 5. Experimental Evaluation

In this Section, we first evaluate the effectiveness of SETUPAGENT for dataset creation and then analyze Code Agent performance across datasets.

### 5.1. Experimental Setup

**Models** We consider a range of models across sizes, cost points, and model providers. For exact versions, see Table 9 in App. A. Unless otherwise specified, we use GPT-4O-MINI as the underlying model for all agents. For decoding, we use the default parameters for all Code Agents and greedy decoding for SETUPAGENT.

**Code Agents** We evaluate three state-of-the-art Code Agents from the top of the SWE-Bench leaderboard[2] which most likely have been optimized for SWE-Bench (Open-Hands (Wang et al., 2024b), AutoCodeRover-v2.0 (Zhang et al., 2024)), and SWE-Agent v1 (Yang et al., 2024a) and ZeroShot (Jimenez et al., 2024) with oracle context (files modified in the ground truth fix) and BM25 retrieval which prompts LLMs directly without any optimization for SWE-Bench. We report the portion of resolved instances as accuracy (Acc.) for all Code Agents.

**Code Execution** We run all code execution (both for SE-TUPAGENT and all Code Agents) in separate Docker containers to improve reproducibility and security. For SETU-PAGENT, we use an Ubuntu 22.04 container as the base image and pre-install a range of common build dependencies but do not provide any Python dependencies.

### 5.2. Effectiveness of SETUPAGENT

We evaluate the effectiveness of SETUPAGENT in creating SWA- and SWEE-Bench by analyzing the frequency of fully successful environment and testing setups in Table 3. We observe SETUPAGENT is able to extract historically correct execution environments for 20-30% of repositories without reference commands and for 55-75% of instances

---

[2]swebench.com accessed in November 2024

*Table 3.* SETUPAGENT success rates at extracting installation and test commands as well as parsing the resulting test output.

|  |  | Success |
|---|---|---|
| SWA | Repos | 28.6% |
|  | Instances | 58.5% |
| SWEE | Repos | 21.6% |
|  | Instances | 71.5% |

*Table 4.* Ablation study on SETUPAGENT, reporting the number of successfully extracted execution environments for SWA-Bench.

|  | # Repositories |
|---|---|
| SETUPAGENT | 44 |
| only CI/CD Files | 33 |
| only Text Files | 15 |
| no Iterative Improvement | 11 |

for these repositories. Without reference commands, SE-TUPAGENT takes 76 minutes to attempt to install all 154 repositories considered for SWA after deduplication and license checks and thus takes only about 30s on average per repository. When creating SWEE-Bench, we deactivate the web browsing ability of SETUPAGENT.

**Ablation**   We evaluate the impact of SETUPAGENT's components in an ablation study on SWA-Bench, reporting results in Table 4. We observe that especially the use of CI/CD config files and the iterative improvement are crucial for SETUPAGENT's success.

**Failure Analaysis**   To understand SETUPAGENT's failure cases, we conduct a small case study, manually inspecting five failed instances from SWA-Bench, and observe the following: In all instances, errors in the build process cause the failure. For all but one instance, finding the installation instructions requires following two or more links on web pages. In all but two instances, the only described way to test the application requires running docker containers, which SETUPAGENT does not support. In two instances, installation and/or testing requires the use of makefiles, referencing multiple substeps. Finally, in one instance SETUPAGENT chooses the wrong requirement file and then begins to install missing testing dependencies. We believe this points to exciting future work improving SETUPAGENT's web-browsing capabilities and docker support.

### 5.3. Agent Performance Across Datasets

We conduct all below experiments on the full SWA and uniformly subsampled versions of SWEE and SWE-Full of identical size (535 instances) due to cost constraints.

We report Code Agent performance in Table 5 and observe

*Table 5.* Issue resolution rates (accuracy) of various agents on SWA-, SWEE-, and SWE-Bench, all with GPT-4O-MINI.

|  | SWA | SWEE | SWE |
|---|---|---|---|
| Openhands | 3.9% | 4.4% | 4.6% |
| AutoCodeRover v2 | 8.4% | 9.0% | 8.2% |
| SWE-Agent v1 | 2.6% | 7.5% | 7.1% |
| ZeroShot(Oracle) | 0.9% | 2.2% | 2.8% |
| ZeroShot(BM25) | 1.3% | 2.8% | 1.5% |

surprisingly small differences in performance between all three datasets when using GPT-4O-MINI for most agents, with SWE-Agent performing significantly worse on SWA.

To assess the interaction of agent performance and model selection, we evaluate AutoCodeRover v2 (Zhang et al., 2024) across a range of LLMs, showing results in Table 6. Interestingly, we observe a large variance in the accuracy difference between SWE and SWA across models. While GPT-4O-MINI performs similarly well on all benchmarks, all other models perform much better on SWEE- and even better on SWE-Bench. We show later that this may be due to lower performances for instances created after the models knowledge cutoff.

*Table 6.* Performance of AutocodeRover v2 (Zhang et al., 2024) using different underlying LLMs.

|  | SWA | SWEE | SWE |
|---|---|---|---|
| GPT-4O-MINI | 8.4% | 9.0% | 8.2% |
| GPT-4O | 10.2% | 15.1% | 16.6% |
| HAIKU-3.5 | 10.8% | 12.9% | 13.6% |
| LLAMA 3.3 70B | 8.8% | 10.8% | 12.5% |
| QWEN2.5 [†] | 3% | 2% | 4% |
| DEEPSEEK V3 [†] | 8% | 13% | 26% |

[†] Evaluated on 100 random instances.

### 5.4. Benchmark Analaysis

In Section 4, we observed interesting distributional differences between the instance characteristics of SWA-, SWEE-, and SWE-Bench. Now, we explore how these characteristics correlate with agent performance, reporting Spearman's rank correlation coefficients $\rho$ and p-values for AutoCodeRover v2 and GPT-4O in Table 7. We observe that only characteristics computed with knowledge of the solution have a statistically significant correlation with performance. In particular, the overlap of the issue with the reference code patch in terms of file names, and number of lines has a strong positive correlation with performance, while all fix complexity metrics have a strong negative correlation with performance.

**Data Contamination**   We analyze the accuracy (Acc) of AutoCodeRover v2 on SWA- and SWEE-Bench, depending on whether a PR was created before or after a model's knowledge cutoff (KC), showing results in Table 8. We re-

*Table 7.* Spearman's rank correlation coefficients $\rho$ and p-value between accuracy and instance characteristics, separated by whether statistic can be computed without axes to ground truth. Statistically significant ($p < 1\%$) correlations are highlighted in bold. Positive $\rho$ indicate that a larger characteristic value is associated with better performance.

| Characteristic | SWA | | SWEE | | SWE | |
|---|---|---|---|---|---|---|
| | $\rho$ | p-value | $\rho$ | p-value | $\rho$ | p-value |
| Repo Age | -0.06 | $2.0 \times 10^{-1}$ | -0.02 | $5.8 \times 10^{-1}$ | -0.02 | $7.0 \times 10^{-1}$ |
| # GitHub Stars | -0.03 | $4.8 \times 10^{-1}$ | -0.02 | $7.2 \times 10^{-1}$ | 0.07 | $1.3 \times 10^{-1}$ |
| # Words in Issue | -0.06 | $1.8 \times 10^{-1}$ | 0.00 | $9.6 \times 10^{-1}$ | 0.01 | $7.7 \times 10^{-1}$ |
| # Code Blocks in Issue | 0.00 | $9.7 \times 10^{-1}$ | 0.04 | $3.4 \times 10^{-1}$ | -0.06 | $1.7 \times 10^{-1}$ |
| # Error Messages in Issue | 0.03 | $4.6 \times 10^{-1}$ | 0.09 | $3.7 \times 10^{-2}$ | -0.04 | $3.5 \times 10^{-1}$ |
| # Fix File Names in Issue | **0.12** | $4.5 \times 10^{-3}$ | **0.19** | $1.2 \times 10^{-5}$ | **0.18** | $2.5 \times 10^{-5}$ |
| Longest Fix Substring in Issue | -0.04 | $3.7 \times 10^{-1}$ | -0.11 | $1.1 \times 10^{-2}$ | 0.04 | $3.1 \times 10^{-1}$ |
| # Fix Lines in Issue | 0.09 | $3.7 \times 10^{-2}$ | 0.06 | $1.7 \times 10^{-1}$ | **0.17** | $1.1 \times 10^{-4}$ |
| # Lines in Fix | **-0.28** | $5.0 \times 10^{-11}$ | **-0.40** | $1.3 \times 10^{-21}$ | **-0.28** | $6.2 \times 10^{-11}$ |
| # Files in Fix | **-0.12** | $6.2 \times 10^{-3}$ | **-0.26** | $1.6 \times 10^{-9}$ | **-0.16** | $1.4 \times 10^{-4}$ |
| # Affected Tests | **-0.18** | $3.3 \times 10^{-5}$ | **-0.25** | $4.0 \times 10^{-9}$ | **-0.15** | $7.2 \times 10^{-4}$ |

*Table 8.* Accuracy of AutocodeRover v2 (Zhang et al., 2024) on SWA-Bench instances split between those created before and after the model's knowledge cutoff (KC) and the p-value of the underlying resolution rate being the same or higher after the KC.

| Dataset | Model | # after KC | Acc before KC | Acc after KC | p-value |
|---|---|---|---|---|---|
| SWA | GPT-4O-MINI | 249 | 9.4% | 7.2% | 17.90% |
| | GPT-4O | 249 | 12.2% | 7.2% | 2.65% |
| | HAIKU-3.5 | 44 | 11.0% | 9.1% | 34.83% |
| SWEE | GPT-4O-MINI | 230 | 8.2% | 10.0% | 76.50% |
| | GPT-4O | 230 | 15.4% | 15.2% | 47.56% |
| | HAIKU-3.5 | 102 | 13.6% | 9.8% | 15.01% |

port the (one-sided) p-value of observing these results under the null hypothesis that the success rate is not lower after the KC (computed using a t-test and normal approximation of the binomial distribution). We observe that on SWA-Bench all considered models have a lower success rate after the KC with the difference being statistically significant only for GPT-4O. Interestingly, we observe no such signs on SWEE-Bench which contains much less popular projects and is thus less prone to contamination. While all SWE instances are too old to conduct a similar analysis, we observe that the performance delta between SWE and SWA is correlated with the drop in accuracy over the KC on SWA.

## 6. Conclusion

We introduced SETUPAGENT, the first method for automated and historically accurate execution environment setup for Python codebases. SETUPAGENT enables us to create repository-level code benchmarks fully automatically from a list of GitHub repositories. We demonstrated its effectiveness by creating two new benchmarks, SWA- and SWEE-Bench, focusing on applications and diversity of codebases, respectively, and addressing several limitations of existing repository-level code benchmarks. In particular, their automated generation allows us to consider many more repositories, increasing diversity and reducing the risk of overfitting, and update the benchmarks over time, minimizing the risk of data contamination.

We extensively analyzed SWA- and SWEE-Bench, observing significant distributional differences compared to SWE-Bench in fix-complexity characteristics that are strongly correlated with agent success. We further found statistically significant performance degradation for SWA-Bench instances created after the knowledge cutoff for one model. Together, these findings highlight the importance of evaluating on diverse, representative, and frequently updated benchmarks and thus the value of our automated benchmark generation approach. We believe SETUPAGENT can facilitate this by enabling practitioners to quickly turn their specific target domain into a high-quality representative benchmark.

## Impact Statement

This paper presents work advancing Code Agent evaluation and may thus amplify all positive and negative societal impacts of improved Code Agents. Our work shows that evaluating Code Agents on diverse and up-to-date benchmarks is critical to obtain representative results, with SWE-Bench (and even more SWE -VERIFIED) consisting of unusually easy problems. These findings may relativize some recent predictions of code agents soon replacing human software developers and show that the field is still far from achieving this goal. Beyond this, our work on automated execution environment setup has the goal of advancing the field of Machine Learning for code more generally. There are many potential additional societal consequences of our work, none of which we feel must be specifically highlighted here.

## Acknowledgements

We would like to thank the anonymous reviewers for their valuable feedback and suggestions, which helped improve the quality of this paper.

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

## A. Appendix: Experiments

Below, we provide the exact model versions we used in Table 9.

*Table 9.* LLM Details inlcuding Knowledge Cutoff (KC)

| Model Name | Model ID | API Provider | KC | Reference |
|---|---|---|---|---|
| GPT-4O | `gpt-4o-2024-08-06` | OpenAI | Oct 2023 | OpenAI (2025) |
| GPT-4O-MINI | `gpt-4o-mini-2024-07-18` | OpenAI | Oct 2023 | OpenAI (2025) |
| HAIKU-3.5 | `claude-3-5-haiku-20241022` | Anthropic | Jul 2024 | Anthropic (2024) |
| DEEPSEEK V3 | `DeepSeek-V3` | TogetherAI | - | Liu et al. (2024) |
| LLAMA 3.3 70B | `Meta-Llama-3.3-70B-Instruct-Turbo` | TogetherAI | Dec 2023 | Grattafiori et al. (2024) |
| QWEN2.5 | `Qwen2.5-72B-Instruct-Turbo` | TogetherAI | | Qwen Team (2024) |

### A.1. Ablations

**Repeatability**   While benchmarks need only be constructed once and thus the repeatability of SETUPAGENT has no impact on the value and repeatability of benchmark's it creates, repeatability is an important aspect for the usability of such a system. Therefore, we conducted an experiment running SETUPAGENT three times on 100 random candidate repositories from SWEE in a setup like for Table 5. We show results in Table 11, where we report the number of successfully installed repositories and the symmetric difference in installed repositories compared to the first run. We find that SETUPAGENT is highly repeatable, with only one repository being installed in the first and second run that was not installed in the third run.

*Table 11.* Repeatability of SETUPAGENT

| Run | Successes | Symmetric Difference to Run 1 |
|---|---|---|
| 1 | 27 | - |
| 2 | 27 | 0 |
| 3 | 26 | 1 |

## B. Appendix: Prompts

In this Section, we provide the full-length prompts used by SETUPAGENT.

*Table 10.* SWA pipeline from projects to tasks. A PR is valid if it resolves an issue, modifies a test file, and is merged. An instance valid, if it has additionally at least one $F \to P$ test.

| Step | # Repos | # PRs |
|---|---|---|
| Initial Projects | 475 | |
| + GH Repo Found | 440 | |
| + Preprocessing | 427 | |
| + Permissive License | 227 | |
| + Has valid PR | 154 | |
| + SETUPAGENT succeeds | 44 | |
| + Get up to 50 valid PRs | | 1527 |
| + SETUPAGENT succeeds | | 893 |
| + valid instance | | 535 |

---

**Prompt to suggest relevant files**

```
You are a senior developer contributing to the www.github.com/<repo_id>
project by solving issues.  You have created a Docker environment with
Ubuntu, and now you want to install the repository in development mode
(meant for active development and testing) and run the tests.  The first
step is to locate the installation instructions and the test commands.
I will provide you a list of filenames or file paths (e.g., README.md,
contributing.md), which typically include instructions for installation
and testing.  The files can be either filenames (e.g., README.md) or file
paths (e.g., docs/maintaining/installing/install-from-source.rst).  From the
provided list of filenames or file paths your task is: 1.  Identify those
likely related to installation or testing based on their names.  2.  Exclude
those that are clearly irrelevant.  3.  If unsure, include the file/path in
your response.  4.  Return only the files/paths from the given list, exactly
as they appear, without modifying their names or structure 5.  If a full
path is given, return the full path, not just the filename.  6.  Use the
following format for your response <ANSWER>:  file 1, ...file n, filepath 1,
...filepath k
<REASONING>:  <YOUR REASONING>
Example input:
```
readme.md, contributing.md, contributors.md,
docs/maintaining/installing/install-from-source.rst,
docs/source/lib/install_datatypes.rst,
docs/html/ux-research-design/contribute.md
```
A reasonable output is:
```
<ANSWER>:  readme.md, contributing.md,
docs/maintaining/installing/install-from-source.rst,
<REASONING>:  The files readme.md and contributing.md
commonly contain installation and testing instructions, while
docs/maintaining/installing/install-from-source.rst is likely related to
installation as the name suggests
```
Here are the file names
```
<file 1>, <file 2>, ..., <file k>
```
Please read the names carefully, ask yourself the purpose of each file based
on the name before including it in your response.  Use the given format for
your answer and please do not add any extra comment or text.
```

*Figure 9.* Prompt for choosing relevant files to installation and testing

**Prompt to suggest external sources of information**

```
You are a senior developer contributing to the GitHub project at
www.github.com/<repo_id> by solving issues.  Your goal is to install the
repository in development mode and run its tests.
You have created a Docker environment with Ubuntu, and now you are searching
for the installation instructions and test commands.
I will provide you with the content of common repository files (e.g.,
README.md, CONTRIBUTING.md).  Your task is to analyze the provided text
and identify all external links that contain relevant information to
1.  Installation instructions for this project.
2.  Test commands or instructions for running the tests for this project.
3.  Contribution guidelines.

Please provide the links you found following the criteria below.
a.  Exclude links to generalpurpose documentation for external tools (e.g.,
Tox, Pytest, or other frameworks/libraries).
b.  If you are unsure about the relevance of a link, better include it.
c.  Order the links from most to least relevant.
d.  Do not add any comment or text.
e.  Use the following format:
LINK: <LINK 1>
LINK: <LINK 2>
...LINK: <LINK n>
Here is the text:
'''
<text_content>
'''
```

*Figure 10.* Prompt to suggest potentially relevant external sources

**Prompt to determine importance of a url content**

```
You are a senior developer working on the GitHub project at
www.github.com/<repo_id>.  You have set up a Docker environment with Ubuntu,
and now your goal is to install the repository in development mode and run
its tests.
Your task is to carefully review the content of the following link:
<current_link>, and determine if it includes installation instructions or
test commands for the <repo_id> project.
Please follow these steps:
1.  Look carefully in the provided content for any potential installation
commands or test commands related to the <repo_id> project.
2.  Ask yourself if the located instructions are reasonable, legitimate
and can be practically executed to install or to test the <repo_id> project
only.
Please provide your answer using the following format:
INSTALLATION/TEST COMMANDS: <TRUE|FALSE>
REASONING: <REASONING>
**Important Notes**
 Answer with TRUE only if the content explicitly includes valid and usable
installation or test commands.
  If you do not find any relevant commands, or if the instructions are
vague, ambiguous, impractical, or unrelated answer FALSE.
 When in doubt, answer FALSE.
Content of the link <current_link>:
'''
<clean_content> '''
```

*Figure 11.* Prompt for determining if a link is relevant to installation and testing in the extraction phase of the SETUPAGENT

**Extract Install Command Prompt**

```
You are a senior developer working on the project located at
www.github.com/<repo_id>.  You have created a Docker environment with
Ubuntu, cloned the repository, and navigated to the directory <repo_dir>.
Your next step is to install the project in development mode, which is
intended for active development and testing.  I'll provide you with
important text files (e.g., README.md) and important continuous integration
(CI) configuration files, which typically contain instructions for
developers on installation and testing.  The format provided will be the
file name followed by its content.
Your task is to identify and return the bash commands necessary for the
correct installation of the repository.  This includes system dependencies,
project installation in development mode, and any prerequisites or
configuration commands.

** IMPORTANT NOTES **
1.  Include system dependencies installation commands required for the
project (e.g., via apt, yum, curl, etc.).
2.  Include installation commands necessary for setting up the project in
development mode.
3.  Include prerequisites installation and configuration commands, such as
those for npm or any other required setup.
3.  If comprehensive installation instructions are provided, return them
without any modifications.
4.  Only exclude commands related to creating or activating virtual
environments.

The returned commands should meet the following criteria:
1.  Enclosed in quotes.
2.  Focused strictly on commands necessary for both system dependency
installation and development-mode installation of the project.
3.  Free from any comments or text.
4.  Accurate and executable without errors.

If no installation commands are present, return NONE.
Here is the text:
```
<context>
```
Take your time to carefully analyze the content.  Make sure that your
response includes only the necessary installation bash commands.  Ask
yourself if the provided content is sufficient for installation.  And for
each command, ask yourself what's the purpose of the command and if it is
necessary.
An example of the expected response is:
```bash
install_command_1
install_command_2
```
Please provide the installation commands in the above specified format.
```

*Figure 12.* Prompt used for extraction of installation commands in extraction phase of SETUPAGENT

**Extract Test Command Prompt**

```
You are a senior developer working on the www.github.com/<repo_id> project.
You have created a Docker environment with Ubuntu, cloned the repository,
and installed it in development mode (meant for active development and
testing).
You are now inside the <repo_dir> directory and your next goal is to run
the unit tests.  I will provide you with some important text files (e.g.,
README.md) and important continuous integration (CI) congiguration files,
which typically include instructions for running tests.  The format provided
will be the file name followed by its content.
Your task is to identify and return the exact bash commands required to run
the tests.
The returned commands should meet the following criteria:
1.  Enclosed in quotes.
2.  Free from any comments or text.
3.  Accurate and executable without errors.
If no test commands are present, return NONE.
Here is the text:
```
<context>
```
Take your time to analyze the content carefully.  Ensure that only the
necessary bash commands for running the tests are included.  Ask yourself
the purpose of each command before including it in your response.
An example of the expected response is:
```bash
test_command_1
test_command_2
```
Please provide the test commands in the above specified format.
```

*Figure 13.* Prompt used for extraction of test commands in the extraction phase of SETUPAGENT

**Prompt for determining error causes**

```
You are a developer working on the project at www.github.com/<repo_id>.  You
created an environment with python version <python_version>.  Your goal is
to install the repository in development mode (meant for active development
and testing) and run the unit tests.
The installation commands are:
```bash
<install_command_1>
<install_command_2>
...
<install_command_k>
```
The testing commands are:
```bash
<test_command_1>
<test_command_2>
...
<test_command_k>
```
You received the following error message after executing the command
<error_command>:
'''
<error_message>
'''
Your task is to analyze the error message and determine its causes.
You can return one of the following answers:
1.  <PYTHON>, if the error is caused by incompatibilities between the python
version and any used package.
2.  <INSTALLATION>, if the error is caused by an installation command or is
related to any missing package, regardless if it a testing related framework
or not.  All the required packages must be installed in the installation
phase.
3.  <TESTING>, if the error is caused by any testing command (e.g., an
invalid flag in the test command)
4.  <UNDECIDABLE>, if you cannot determine what causes the error.
Please read the error message carefully and try to spot the commands that
are responsible for the error.  Always provide the reasoning for your
answer.
Use the following format:
RESULT: <PYTHON, INSTALLATION, TESTING, UNDECIDABLE>
REASONING: <YOUR REASONING>
```

*Figure 14.* Prompt for determining the error cause in the iterative improvement phase of the SETUPAGENT

---

**Prompt for fixing python version**

```
You are a senior developer working on the project at
www.github.com/<repo_id>.  Your goal is to install the repository in
development mode (meant for active development and testing) and run the
unit tests.
You created an environment with python version <python_version>, but you are
unsure if the python version is correct.
You received the following error message while testing the repository:
'''
<error_message>
'''
A senior software developer colleague has provided an explanation of why
things are not working as expected with the current commands:
<Reasoning from the answer to the prompt for determining the error cause>.
Use his reasoning to resolve the current error we are facing.
Your task is to determine a compatible Python version for the current state
of the repository.  Carefully read the error message and identify the most
suitable Python version.
Please follow this answer format:
1.  Return <NONE> if the error is unrelated to the Python version or you
cannot determine a compatible version.
2.  If a specific Python version is compatible, return only the version
number (e.g., 2.7).
3.  Do not include any additional comments or text in your response.
```

*Figure 15.* Prompt for fixing python version used in the iterative improvement phase of SETUPAGENT

---

**Prompt for fixing installation commands 1**

```
You are a senior developer working on the project at
www.github.com/<repo_id>.  You are working in an enviroment with python
version <python_version>.  You have attempted to install the repository
in development mode (meant for active development and testing) using the
following bash commands:
```bash
<install_command_1>
<install_command_2>
...
<install_command_n>
```
However, the command <error_command> failed and we received the following
error message:
'''
<error_message> '''
Your task is to fix the above error.  Think carefully what causes the error
and try to spot the commands that are responsible for it.  Please provide
the updated installation steps in a bash code block, following these rules:
1.  You have to use always uv pip instead of regular pip.
2.  Return <NONE> if you can not fix the command.
3.  Do not add any comments or text.
For example:
```bash
apt-get install -y <package_name>
uv pip install -r requirements.txt
```
```

*Figure 16.* Prompt for fixing the installation commands used in the iterative improvement phase of SETUPAGENT when the error occurs in the building process of containers

**Prompt for fixing installation commands 2**

You are a senior developer working on the project at
www.github.com/<repo_id>.  You tried to install the repository in
development mode, which is intended for active development and testing,
however the installation failed.
You are working in an enviroment with python version <python> and you tried
to use the following bash commands for the installation:
```bash
<install_command_1>
<install_command_2>
...
<install_command_n>
```
During the execution of these commands, you received the following error
message:  '''
<error_message> '''
A senior software developer colleague has provided an explanation of why
things are not working as expected with the current commands:
<Reasoning from the answer to the prompt for determining the error cause>.
Use his reasoning to resolve the current error we are facing.
Your task is to carefully read the error message and determine which
commands are causing the error.  Reason about every command if it is causing
the error.  If you conclude that the problem is related to any of the
commands, update the installation bash script to solve the problem.  Note
that you can also add new commands to fix the problem.  If you decide to
update the installation bash script you have to follow these rules:
1.  Provide the updated installation steps in a bash code block.
2.  Use uv pip instead of regular pip.
2.  Return NONE if the error is not related to the installation steps or you
are not able to fix it.
3.  Do not add any comments or text.
For example:
The initial installation command is:
```bash
uv pip install ˙
```
However, the error message states that the <package_name> package is not
installed.  Then you would update the installation command to:
```bash
uv pip install ˙
uv pip install <package_name> ```

*Figure 17.* Prompt for fixing the installation commands used in the iterative improvement phase of SETUPAGENT

**Prompt for fixing testing commands**

```
You are a senior developer working on the project at
www.github.com/<repo_id>.  You installed the repository in an enviroment
with python version <python_version> and now you are trying to run the unit
tests.
You run the tests using the following bash commands:
```bash
<test_command_1>
<test_command_2>
...
<test_command_k>
```
However, at the moment we receive the following error message:
'''  <error_message> '''
A senior software developer colleague has provided an explanation of why
things are not working as expected with the current commands:
<Reasoning from the answer to the prompt for determining the error cause>.
Use his reasoning to resolve the current error we are facing.
Your task is to read the produced error message carefully, determine what
the problem is and try to fix it.  Ask yourself which test command could
cause this problem.  If you conclude that the problem is related to the test
commands, update the test commands to solve the problem.
Please provide the updated test commnds in a bash code block, following
these rules:
1.  You have to always use uv pip instead of regular pip.
2.  Return NONE if the error is not related to the test command or you
cannot fix it.
3.  Do not add any comments or text.
4.  Add a command only if you are sure that it is correct.
For example:  The initial testing command was:
```bash
pytest test_file.py run all ``` However, if in this case we would need the
flag '-v' and the maximal number of failing tests to be 1, we would have to
correct the command to:
```bash
pytest test_file.py maxfail=1 v ```
```

*Figure 18.* Prompt for fixing the installation commands used in the iterative improvement phase of SETUPAGENT

# C. Appendix – Dataset Details

Below, we list all repositories along with the number of corresponding tasks in SWA-Bench.

**SWA-Bench– Repositories**

 1. iterative/dvc – 42
 2. streamlink/streamlink – 35
 3. spack/spack – 35
 4. PrefectHQ/prefect – 34
 5. xonsh/xonsh – 32
 6. mitmproxy/mitmproxy – 31
 7. python-pillow/Pillow – 29
 8. mkdocs/mkdocs – 23
 9. hynek/structlog – 22
10. pallets/click – 21
11. locustio/locust – 20
12. jpadilla/pyjwt – 17
13. elastic/elasticsearch-dsl-py – 17
14. pallets-eco/wtforms – 17
15. ipython/ipython – 16
16. python-poetry/poetry – 15
17. conan-io/conan – 15
18. sabnzbd/sabnzbd – 14
19. Zulko/moviepy – 14
20. nvbn/thefuck – 12
21. arrow-py/arrow – 11
22. benoitc/gunicorn – 8
23. cookiecutter/cookiecutter – 8
24. pypa/pipenv – 7
25. graphql-python/graphene – 6
26. pypa/bandersnatch – 5
27. AtsushiSakai/PythonRobotics – 4
28. hynek/doc2dash – 3
29. PythonCharmers/python-future – 3
30. aimhubio/aim – 2
31. dbcli/pgcli – 2
32. geopython/pycsw – 2
33. dbader/schedule – 2
34. kibitzr/kibitzr – 1
35. getnikola/nikola – 1
36. geopy/geopy – 1
37. Maratyszcza/PeachPy – 1
38. gawel/pyquery – 1
39. Suor/funcy – 1
40. simonw/datasette – 1
41. cowrie/cowrie – 1
42. pypa/pip – 1
43. StevenBlack/hosts – 1
44. jupyter/nbgrader – 1

Below, we list all repositories along with the number of corresponding tasks in SWEE-Bench.

## SWEE-Bench– Repositories Part I

1. python-attrs/attrs – 9
2. dgasmith/opt_einsum – 9
3. jazzband/tablib – 8
4. MartinThoma/flake8-simplify – 8
5. matthewwithanm/python-markdownify – 8
6. stephenhillier/starlette_exporter – 8
7. sciunto-org/python-bibtexparser – 8
8. davidhalter/parso – 8
9. marshmallow-code/flask-smorest – 7
10. adamchainz/blacken-docs – 7
11. MarketSquare/robotframework-tidy – 7
12. lundberg/respx – 7
13. seperman/deepdiff – 7
14. Stranger6667/hypothesis-graphql – 7
15. cantools/cantools – 7
16. didix21/mdutils – 7
17. marshmallow-code/apispec – 7
18. softlayer/softlayer-python – 6
19. gorakhargosh/watchdog – 6
20. pygments/pygments – 6
21. dask-contrib/dask-histogram – 6
22. andialbrecht/sqlparse – 6
23. mirumee/ariadne – 6
24. tableau/tabcmd – 6
25. gerrymanoim/exchange_calendars – 5
26. snowplow/snowplow-python-tracker – 5
27. joerick/pyinstrument – 5
28. scikit-rf/scikit-rf – 5
29. matthewwardrop/formulaic – 5
30. laspy/laspy – 5
31. python-control/python-control – 5
32. mwouts/itables – 5
33. AzureAD/microsoft-authentication-library-for-python – 5
34. firebase/firebase-admin-python – 5
35. ethereum/eth-account – 5
36. davidhalter/jedi – 5
37. agronholm/typeguard – 5
38. Delgan/loguru – 5
39. pytransitions/transitions – 5
40. lovasoa/marshmallow_dataclass – 5
41. aio-libs/yarl – 5
42. PyCQA/pyflakes – 5
43. python/importlib_metadata – 5
44. konradhalas/dacite – 5
45. ilevkivskyi/typing_inspect – 5
46. jupyter/jupyter_core – 5
47. getsentry/responses – 5
48. beartype/plum – 4
49. open2c/bioframe – 4
50. developmentseed/morecantile – 4
51. nats-io/nats.py – 4
52. nipy/nipype – 4
53. python-quantities/python-quantities – 4
54. stac-utils/pystac-client – 4
55. luolingchun/flask-openapi3 – 4
56. sayanarijit/expandvars – 4
57. jpadilla/pyjwt – 4
58. NowanIlfideme/pydantic-yaml – 4
59. john-kurkowski/tldextract – 4
60. geopandas/geopandas – 4
61. cloudevents/sdk-python – 4
62. jupyter/nbformat – 4
63. matthew-brett/delocate – 4
64. iterative/shtab – 4
65. jsonpickle/jsonpickle – 4
66. ethereum/eth-utils – 4
67. mhe/pynrrd – 4
68. adamjstewart/fiscalyear – 4
69. pytest-dev/pytest-xdist – 4
70. facelessuser/wcmatch – 4
71. scikit-hep/awkward – 4
72. tomplus/kubernetes_asyncio – 4
73. ipython/traitlets – 4
74. David-Wobrock/sqlvalidator – 4
75. omry/omegaconf – 4
76. python-lsp/python-lsp-server – 4
77. cogeotiff/rio-tiler – 3
78. wjohnson/pyapacheatlas – 3

## SWEE-Bench – Repositories Part II

79. adamchainz/django-htmx – 3

80. mwclient/mwclient – 3

81. executablebooks/sphinx-book-theme – 3

82. scikit-hep/vector – 3

83. patrick-kidger/equinox – 3

84. christiansandberg/canopen – 3

85. regebro/pyroma – 3

86. nephila/giturlparse – 3

87. cookiecutter/cookiecutter – 3

88. serge-sans-paille/pythran – 3

89. tomasvotava/fastapi-sso – 3

90. jsvine/pdfplumber – 3

91. scrapy/protego – 3

92. SmileyChris/django-countries – 3

93. cscorley/whatthepatch – 3

94. pythological/kanren – 3

95. pypa/virtualenv – 3

96. fastavro/fastavro – 3

97. marshmallow-code/marshmallow-sqlalchemy – 3

98. gazpachoking/jsonref – 3

99. lepture/mistune – 3

100. scikit-learn-contrib/category_encoders – 3

101. simonw/sqlite-utils – 3

102. executablebooks/mdit-py-plugins – 3

103. tsutsu3/linkify-it-py – 3

104. hhatto/autopep8 – 3

105. cubewise-code/mdxpy – 3

106. joblib/joblib – 3

107. python-trio/trio-typing – 3

108. nalepae/pandarallel – 3

109. tableau/server-client-python – 3

110. r1chardj0n3s/parse – 3

111. ipython/ipython – 3

112. pypa/readme_renderer – 3

113. jaraco/zipp – 3

114. docker/docker-py – 3

115. joshy/striprtf – 3

116. googleapis/python-pubsub – 3

117. TylerYep/torchinfo – 3

118. scrapy/w3lib – 3

119. googleapis/google-auth-library-python-oauthlib – 3

120. agronholm/cbor2 – 3

121. weiwei/junitparser – 3

122. conan-io/conan – 3

123. python/importlib_resources – 3

124. timvink/mkdocs-git-authors-plugin – 3

125. agronholm/exceptiongroup – 3

126. magmax/python-inquirer – 3

127. PrefectHQ/prefect – 3

128. Yelp/detect-secrets – 3

129. Chilipp/autodocsumm – 3

130. jaraco/keyring – 3

131. Pylons/waitress – 3

132. pypa/setuptools – 3

133. barrust/pyspellchecker – 2

134. bluesky/ophyd – 2

135. OpenMath/py-openmath – 2

136. readthedocs/sphinx-notfound-page – 2

137. canonical/operator – 2

138. ekzhu/datasketch – 2

139. dhatim/python-license-check – 2

140. Shoobx/xmldiff – 2

141. ewels/rich-click – 2

142. jaraco/path – 2

143. yu-iskw/dbt-artifacts-parser – 2

144. symerio/pgeocode – 2

145. daggaz/json-stream – 2

146. jazzband/dj-database-url – 2

147. nipunsadvilkar/pySBD – 2

148. adamchainz/django-linear-migrations – 2

149. mwouts/jupytext – 2

150. MrBin99/django-vite – 2

151. ml31415/numpy-groupies – 2

152. regebro/svg.path – 2

153. gmr/flatdict – 2

154. aws-samples/sample-python-helper-aws-appconfig – 2

155. behave/behave – 2

## SWEE-Bench – Repositories Part III

```
156. thesimj/envyaml – 2
157. codingjoe/django-select2 – 2
158. allisson/python-simple-rest-client – 2
159. christianhelle/autofaker – 2
160. esphome/aioesphomeapi – 2
161. oauthlib/oauthlib – 2
162. rustedpy/result – 2
163. graphql-python/graphene – 2
164. benmoran56/esper – 2
165. eerimoq/bincopy – 2
166. keleshev/schema – 2
167. PyCQA/flake8 – 2
168. kjd/idna – 2
169. jupyter/nbconvert – 2
170. scikit-hep/hist – 2
171. spulec/freezegun – 2
172. jupyter/nbclient – 2
173. PythonCharmers/python-future – 2
174. tortoise/pypika-tortoise – 2
175. rthalley/dnspython – 2
176. mkaranasou/pyaml_env – 2
177. terraform-compliance/cli – 2
178. googleapis/python-firestore – 2
179. googleapis/python-api-core – 2
180. scrapy/cssselect – 2
181. python-humanize/humanize – 2
182. jdepoix/youtube-transcript-api – 2
183. dedupeio/dedupe – 2
184. databricks/databricks-cli – 2
185. bluesky/event-model – 2
186. workos/workos-python – 2
187. kynan/nbstripout – 2
188. assertpy/assertpy – 2
189. dbt-labs/hologram – 2
190. sendgrid/python-http-client – 2
191. keis/base58 – 2
192. attwad/python-osc – 2
193. wireservice/csvkit – 2
194. adamchainz/time-machine – 2
195. MagicStack/immutables – 2
```

```
196. vinitkumar/json2xml – 2
197. frispete/keyrings.cryptfile – 2
198. swansonk14/typed-argument-parser – 2
199. scottwernervt/favicon – 2
200. slackapi/python-slack-sdk – 2
201. nginxinc/crossplane – 2
202. hetznercloud/hcloud-python – 2
203. dbader/schedule – 2
204. amplify-education/python-hcl2 – 2
205. jazzband/contextlib2 – 2
206. theskumar/python-dotenv – 2
207. raimon49/pip-licenses – 2
208. locustio/locust – 2
209. astanin/python-tabulate – 2
210. alecthomas/voluptuous – 2
211. django-crispy-forms/crispy-bootstrap5 – 2
212. geospace-code/pymap3d – 2
213. tedder/requests-aws4auth – 2
214. pyvisa/pyvisa-py – 1
215. nithinmurali/pygsheets – 1
216. mlenzen/collections-extended – 1
217. emcconville/wand – 1
218. rsalmei/alive-progress – 1
219. rycus86/prometheus_flask_exporter – 1
220. fastapi-users/fastapi-users – 1
221. google/mobly – 1
222. scrapy/itemadapter – 1
223. ncclient/ncclient – 1
224. google/duet – 1
225. di/calver – 1
226. beancount/smart_importer – 1
227. bridgecrewio/python-hcl2 – 1
228. construct/construct – 1
229. devrimcavusoglu/pybboxes – 1
230. richardpenman/whois – 1
231. cvxpy/cvxpy – 1
232. elastic/ecs-logging-python – 1
233. pythonarcade/pytiled_parser – 1
234. astropy/extension-helpers – 1
```

## SWEE-Bench – Repositories Part IV

235. SAP/python-pyodata – 1

236. Azure/azure-functions-durable-python – 1

237. IdentityPython/djangosaml2 – 1

238. jwodder/check-wheel-contents – 1

239. Zulko/moviepy – 1

240. xhtml2pdf/xhtml2pdf – 1

241. cknd/stackprinter – 1

242. guillp/jwskate – 1

243. jmcarp/flask-apispec – 1

244. timofurrer/colorful – 1

245. miso-belica/sumy – 1

246. kvesteri/intervals – 1

247. marcotcr/lime – 1

248. wkentaro/gdown – 1

249. realpython/codetiming – 1

250. jaraco/tempora – 1

251. jendrikseipp/vulture – 1

252. pycontribs/ruyaml – 1

253. albumentations-team/albumentations – 1

254. nose-devs/nose2 – 1

255. jongracecox/anybadge – 1

256. patrys/httmock – 1

257. maxfischer2781/asyncstdlib – 1

258. pgzip/pgzip – 1

259. arvkevi/kneed – 1

260. rasterio/affine – 1

261. circus-tent/circus – 1

262. xchwarze/samsung-tv-ws-api – 1

263. jaraco/portend – 1

264. fabiocaccamo/python-benedict – 1

265. numpy/numpy-financial – 1

266. praw-dev/prawcore – 1

267. scipy/oldest-supported-numpy – 1

268. logtail/logtail-python – 1

269. polkascan/py-scale-codec – 1

270. Knio/pynmea2 – 1

271. jazzband/django-configurations – 1

272. allenai/cached_path – 1

273. click-contrib/click-aliases – 1

274. Pylons/hupper – 1

275. cloudscale-ch/cloudscale-python-sdk – 1

276. alessandromaggio/pythonping – 1

277. imageio/imageio-ffmpeg – 1

278. podhmo/python-node-semver – 1

279. netbox-community/pynetbox – 1

280. kumar303/mohawk – 1

281. SpamScope/mail-parser – 1

282. perrygeo/python-rasterstats – 1

283. pahaz/sshtunnel – 1

284. python-hyper/h11 – 1

285. razorpay/razorpay-python – 1

286. zeroSteiner/rule-engine – 1

287. mocobeta/janome – 1

288. glut23/webvtt-py – 1

289. benoitc/gunicorn – 1

290. mcmtroffaes/pybtex-docutils – 1

291. alexmojaki/executing – 1

292. sigmavirus24/github3.py – 1

293. ccpem/mrcfile – 1

294. csinva/imodels – 1

295. click-contrib/click-help-colors – 1

296. srossross/rpmfile – 1

297. hgrecco/pint – 1

298. django-ses/django-ses – 1

299. gmr/pamqp – 1

300. spotify/annoy – 1

301. PyCQA/pycodestyle – 1

302. regebro/tzlocal – 1

303. mapado/haversine – 1

304. scientific-python/lazy-loader – 1

305. grappa-py/grappa – 1

306. flexmock/flexmock – 1

307. jg-rp/liquid – 1

308. prompt-toolkit/python-prompt-toolkit – 1

309. jaraco/jaraco.context – 1

310. aio-libs/multidict – 1

311. rsheftel/pandas_market_calendars – 1

## SWEE-Bench – Repositories Part V

312. mkdocs/mkdocs – 1

313. websocket-client/websocket-client – 1

314. DataDog/datadog-lambda-python – 1

315. iterative/dvclive – 1

316. cogeotiff/rio-cogeo – 1

317. erikrose/parsimonious – 1

318. facelessuser/pymdown-extensions – 1

319. pypa/build – 1

320. mkdocs/mkdocs-redirects – 1

321. dlint-py/dlint – 1

322. klen/peewee_migrate – 1

323. afq984/python-cxxfilt – 1

324. kinverarity1/lasio – 1

325. Turbo87/utm – 1

326. django/daphne – 1

327. executablebooks/sphinx-design – 1

328. interpretml/slicer – 1

329. google/yapf – 1

330. sensein/etelemetry-client – 1

331. MKuranowski/aiocsv – 1

332. executablebooks/sphinx-tabs – 1

333. pexpect/pexpect – 1

334. pythological/etuples – 1

335. frankie567/httpx-oauth – 1

336. sarugaku/resolvelib – 1

337. python273/telegraph – 1

338. boolangery/py-lua-parser – 1

339. Electrostatics/mmcif_pdbx – 1

340. pyca/service-identity – 1

341. diff-match-patch-python/diff-match-patch – 1

342. xlwings/jsondiff – 1

343. mapbox/cligj – 1

344. cthoyt/pystow – 1

345. Rapptz/discord.py – 1

346. gahjelle/pyplugs – 1

347. Colin-b/pytest_httpx – 1

348. LLNL/certipy – 1

349. spec-first/connexion – 1

350. Yelp/bravado – 1

351. mkorpela/pabot – 1

352. scrapy/parsel – 1

353. alexmojaki/pure_eval – 1

354. graphql-python/graphql-core – 1

355. joke2k/faker – 1

356. averbis/averbis-python-api – 1

357. jupyter/jupyter_client – 1

358. jaraco/inflect – 1

359. GreyZmeem/python-logging-loki – 1

360. suminb/base62 – 1

361. youknowone/wirerope – 1

362. xnuinside/simple-ddl-parser – 1

363. executablebooks/sphinx-thebe – 1

364. Pylons/webob – 1

365. SethMMorton/fastnumbers – 1

366. python-semver/python-semver – 1

