# OpenReview forum: "Automated Benchmark Generation for Repository-Level Coding Tasks"
_ICML.cc/2025/Conference — ICML 2025 poster_

### Official Review · Reviewer_XqFN · 2025-03-17

**Overall Recommendation:** 4

**Summary:**

This works presents SETUPAGENT - an automated LLM-based approach to generate repository-level coding benchmarks from a list of github repos. It enables creation of larger and diverse benchmarks suitable for evaluating software engineering agents, and automates steps such as dependencies setup, test execution, result parsing, etc.

Authors use SETUPAGENT to create 2 new benchmarks: SWEE-Bench (extended version of SWE-Bench encompassing more diverse repositories) and SWA-Bench (focusing on applications rather than libraries).

Authors highlight the distributional differences of SWEE and SWA to SWE-Bench (issue description quality, fix complexity, etc.) and highlight possible contamination of models by showing difference in SWE agent success rates before and after model’s knowledge cutoff dates.

**Claims And Evidence:**

Yes

- The claim of 40% lower agent success rates on SWEE and SWA compared to SWE claim is unclear. Table 5 and 6 show very similar success rates for all 3 benchmarks, only some difference between SWEE and SWA with one of the models.

**Essential References Not Discussed:**

Nil

**Experimental Designs Or Analyses:**

Yes

**Methods And Evaluation Criteria:**

Yes

**Other Comments Or Suggestions:**

None

**Other Strengths And Weaknesses:**

Weakness:
- Authors have shown experiments with 2 models (GPT4o and 4omini), any performance differences of SWE agents using opensource models on SWE, SWA, and SWEE will strengthen the value of the work.
- Although authors point out distributional differences of SWE with SWEE & SWA, a deeper understanding (based on multiple attributes) of type of new diverse samples present in SWEE and SWA (not present in SWE) would be welcome.

**Questions For Authors:**

See claims & weakness sections

**Relation To Broader Scientific Literature:**

Overall, the LLM-based approach to generate SWE benchmarks is novel and very valuable. This work contributes 2 new benchmarks, extending the SWE-Bench & addresses its limitations (diversity, potential contamination of models.).

**Theoretical Claims:**

None

---

> ### Author Rebuttal · Authors · 2025-04-01
>
> We thank the reviewer for their valuable and positive review, highlighting both the novelty of our work and the value it brings to the community. Below, we address their remaining questions:
>
> **Can you extend the evaluation of the created benchmarks?**
> We have added three additional agents/methods (SWE-agent v1, ZeroShot-BM25, ZeroShot-Oracle) and four additional models (Haiku3.5, Llama 3.3 70B, Qwen 2.5 72B, DeepSeek V3) to our evaluation. See the results below
>
> **Model comparison with ACR**
> | |SWA|SWEE|SWE|
> |-|-|-|-|
> |GPT4o-mini| 8.4% | 9.0% | 8.2% |
> |GPT4o| 10.2% | 15.1% | 16.6% |
> |Haiku| 10.8% | 12.9% | 13.6% |
> |Llama3.3 70B| 8.8% | 10.8% | 12.5% |
> |Qwen2.5 72B *| 3% | 2%| 4% |
> |DeepSeek V3 *| 8% | 13% | 26% |
> *only evaluated on a random subset of 100 instances
>
> **Agent comparison with GPT4o-mini**
> | |SWA|SWEE|SWE|
> |-|-|-|-|
> |AutoCodeRover v2| 8.4% | 9.0% | 8.2% |
> |Openhands| 3.9% | 4.4% | 4.6% |
> |SWE-Agent v1|2.6% | 8.9% | 8.2% |
> |ZeroShot (Oracle) | 0.9% | 2.2% | 2.8% |
> |ZeroShot (BM 25) | 1.3% | 2.8% | 1.5% |
>
> While most of the additional results are in line with what we observed so far, we want to highlight that there are some interesting behaviours. In particular, DeepSeek V3 performs much better (2-3x) on SWE than on both other benchmarks, similarly SWE-Agent v1 performs much worse on SWA than on both SWEE and SWE. This highlights the value of diverse benchmarks, enabling a holistic evaluation of new models and methods.
>
> **Where do you observe the performance differences mentioned in the Abstract?**
> E.g., in Table 6, we show that the performance of GPT4o is 40% lower on SWA compared to SWE. We observe even bigger differences in the results reported above.

---

### Official Review · Reviewer_msS6 · 2025-03-22

**Overall Recommendation:** 2

**Summary:**

The paper addresses the problem of automatically creating repository-level execution benchmarks for software engineering tasks. The authors first describe SetupAgent, an LLM-powered agentic framework to setup the execution environment for any Python repository. This is then used to create SWA- and SWEE-Bench, increasing diversity of problems represented and minimizing contamination against models like GPT-4o. Evaluations show similar level of performance on SWE-Bench as these benchmarks with SWA-Bench showing slightly lower performance for problems after the models’ knowledge cut-off.

**Claims And Evidence:**

1. SetupAgent is a very useful contribution to the community allowing researchers to turn any Python repository into a benchmark instance. However, this system is not adequately evaluated for accuracy or reliability. While the authors run tests reproduce original issues, I believe that the system needs to evaluated deeper to check the presence, correctness, reliability and reproducibility of fail-to-pass tests. A thorough evaluation of the system with this in mind is missing from the paper. Further, SetupAgent incorporates multiple LLM-guided steps, and as far as I can tell, no attempts were made to vet the LLM outputs. While the system proposal is useful, I think the authors should take more steps towards improving the transparency of the robustness and reliability of SetupAgent.
2. The paper claims higher diversity in SWEE- and SWA-Bench compared to SWE-Bench. These are backed by statistical studies in Section 4.
3. Finally, the statistical significance of performance of various models on various benchmarks are also shown. In some cases the p-values are very high, e.g., Table 8 has many p-values > 0.1. Nevertheless, I commend the authors for being transparent about these results.

**Essential References Not Discussed:**

I have touched upon this above. The problems raised and addressed in the SWE-Bench Verified work are not mentioned in this work. I think this is essential for the adoption of SetupAgent, SWA- and SWEE-Bench just as SWE-Bench Verified is now the default benchmark for evaluation.

**Experimental Designs Or Analyses:**

I have answered this in other parts of my review.

**Methods And Evaluation Criteria:**

I have specified a brief commentary on the creation of SetupAgent above. Besides that, I think the authors also miss to comment on the merits and comparisons of their efforts to SWE-Bench Verified [1]. It is now known that the original SWE-Bench has many issues at the systemic and problem semantic levels, and SWE-Bench Verified was an effective annotation effort to address such concerns. Given that SetupAgent leads to a similar benchmark as SWE-Bench, I think that the authors should also check if the same concerns are present in SWA- and SWEE-Bench.

[1] https://openai.com/index/introducing-swe-bench-verified/

**Other Comments Or Suggestions:**

Line 375 - e → We

**Other Strengths And Weaknesses:**

1. I believe that SetupAgent solves an important problem of automating the creation of execution environments but this needs to be done in a more rigorous way. I have elaborated on this above.
2. I think that Section 3.1 is completely unnecessary. It obfuscates a pretty simple setup and does not add much value. I recommend the authors to remove this section and all the following mathematical notations.
3. I did not understand the difference between SWA- and SWEE-Bench. Specifically, I’m not sure about what makes SWA-Bench unique.  I do not understand this statement: “Code Agents develop software applications that suffer from different types of bugs compared to libraries due to architectural and structural differences.” Could you elaborate or maybe show an example?
4. I’m not sure if this should be a major or minor concern: I do not see much difference in model performances across SWE- and SWEE-/SWA-Bench (Table 5, 6). Am I missing something from these results?

**Questions For Authors:**

1. Can you concretely clarify the difference between SWA- and SWEE-Bench? Are both needed? If not, which one is better? This will help researchers understand what’s useful for their use case.
2. How would you judge the accuracy and reliability of SetupAgent? Have you evaluated the various components of SetupAgent? This will help increase the trust in the system.
3. What is the status of issues raised in SWE-Bench Verified in SWEE- and SWA-Bench? Did you take any steps to address these issues? If not, what should researchers using your benchmarks be mindful of?

**Relation To Broader Scientific Literature:**

The paper is related to the evaluation of LLMs as agents in software engineering. The most relevant work in this direction is SWE-Bench [1].

[1] Jimenez, Carlos E., et al. "Swe-bench: Can language models resolve real-world github issues?." *arXiv preprint arXiv:2310.06770* (2023).

**Theoretical Claims:**

n/a

---

> ### Author Rebuttal · Authors · 2025-04-01
>
> We thank the reviewer for their detailed review, acknowledging the value of our contribution to the community and the quality of our analysis. Below, we address their remaining concerns.
>
> **Can you evaluate the presence and correctness of fail-to-pass tests more rigorously?**
> We first want to highlight that we already filter the dataset to only include instances where we extracted F2P tests by executing the test suite in the original state with test changes applied and in the fixed state with test changes applied. We have additionally conducted a manual review of the extracted tests (see response to Reviewer FoWC) and find them to be of high quality for 90% of instances.
> Finally, we have rerun the test execution and extraction 10 times for SWA to identify possibly flaky tests and extract 509 shared instances across all 10 runs (compared to 535 from the original run). We traced the remaining discrepancy down to two main causes: randomly parameterized tests and genuinely flaky tests. We will create a version of both the SWA and SWEE datasets with these filtered out but expect this to have minimal impact on our results and note that similar behaviour is present in SWE-Bench (see e.g., https://arxiv.org/pdf/2407.21787).
>
> **Have you evaluated SetupAgent both as a whole and with respect to its individual components?**
> We evaluate SetupAgent in Section 5.2, discussing its overall performance in Table 3 and conducting an ablation study in Table 4, demonstrating the importance of all its components. We have now additionally conducted a human evaluation, discussed in the response to Reviewer FoWC.
>
> **Can you analyse the issues discussed in SWE-Bench-Verified in context of your benchmarks?**
> Given the similarity in the benchmark generation pipeline, we expect similar findings to those for SWE-Bench(-Verified). However, the two main review/filtering criteria in the creation of SWE-Verified, namely, the completeness of the issue description and the specificity of added tests, only aim to filter out instances that can not be solved without additional information. This is crucial to assess the absolute performance of code generation systems but will affect all evaluated systems to a similar extent and thus have only marginal impact on a comparative analysis. We note that such a comparative analysis is sufficient to select among different systems for a given use case or guide their development. Similarly, we focus our analysis on the differences in the relative performance between models and compare our benchmarks to SWE-Full and not SWE-Verified.
> Finally, we have conducted a similar human review as for the creation of SWE-Verified on 30 random samples of SWA, which we discuss in the response to Reviewer FoWC. We are happy to include this discussion in the next revision of our work.
>
> **Can you discuss the differences between SWE, SWA, and SWEE? Are all needed?**
> The main difference is the set of considered projects. While SWE(E) focuses on libraries that typically expose functionality to other programs and thus tend to have more stable, well-tested APIs, SWA focuses on applications (e.g., locust or xonsh) that are directly used by humans and often have (graphical) UIs. As we show in Table 6, the performance of some models (GPT4o-min) is quite consistent across these different tasks, while others (GPT4o) varies significantly. We have conducted additional experiments (see response to Reviewer XqFN) where we see significant (3x) differences across benchmark for SWE-Agent v1 and DeepSeek V3. We believe this demonstrates the value of more diverse benchmarks in this highly active field.
>
> **Comment on Section 3.1**
> We introduced the notation in Section 3.1 to formalize our plain English (and sometimes incomplete for brevity's sake) descriptions of different settings. However, we are happy to move this formalization and its usages to the appendix.
>
> **Conclusion**
> We hope to have addressed the Reviewer's concerns, look forward to their reply, and remain happy to answer follow-up questions.

---

### Official Review · Reviewer_FowC · 2025-03-22

**Overall Recommendation:** 2

**Summary:**

This paper introduces SETUPAGENT, a system for automatically generating repository-level benchmarks for code agents by setting up historically accurate execution environments. It extracts installation and testing commands from GitHub repositories using LLMs, iteratively refines them based on execution feedback, and validates correctness through test suite execution. Using SETUPAGENT, the authors create two new large-scale benchmarks: SWA-Bench, focused on software applications; SWEE-Bench, focused on diverse and less-popular Python repositories.

**Claims And Evidence:**

A few areas could benefit from additional validation.
For example, " SETUPAGENT ensures high correctness and historical fidelity", while the 95% pass-rate threshold and test-level granularity are good proxies, it seems no human validation or reproducibility check.

**Essential References Not Discussed:**

N.A.

**Experimental Designs Or Analyses:**

Benchmark quality relies on automated test results, which may miss subtle correctness issues (e.g., partial fixes, untested behaviors).

No human evaluation or baseline “gold set” is used to further validate the correctness of the generated tasks.

**Methods And Evaluation Criteria:**

- The paper tackles the challenge of limited, manually curated benchmarks like SWE-Bench.
- Integration of LLMs for interpreting semi-structured repo files: test execution provides concrete, verifiable feedback to supervise LLM-generated commands.
- The authors balance automation with correctness through iterative feedback-driven correction and 95% test pass threshold in the validation phase

**Other Comments Or Suggestions:**

Consider randomly sampling a subset (e.g., 50 instances) of generated benchmark tasks for manual review to verify: (1) The issue and patch are meaningfully connected. (2)The extracted test suite is valid and reflects the issue. (3) The task is realistic and unambiguous.
This would boost confidence in the correctness and representativeness of the dataset.

A direct head-to-head comparison (in terms of runtime, accuracy, coverage, and granularity) with EXECUTIONAGENT would solidify SETUPAGENT’s claimed superiority. Even a small-scale test (e.g., on 10 shared repos) would be informative.

Consider including an example of how external researchers can integrate the benchmark into their own code agent evaluation pipelines.

Add more baseline models of SWE-bench to test your benchmarks. (https://www.swebench.com/#verified)

**Other Strengths And Weaknesses:**

Strengths
- The paper goes beyond reporting benchmark creation—it analyzes failure modes, conducts ablation studies, and investigates performance correlations with task characteristics.
- Designed to scale to thousands of benchmark instances without human intervention.
- The system can be reused for future benchmark updates with new repositories or tasks.

Weakness
- No qualitative or manual evaluation is done to assess the correctness or realism of generated benchmark instances. The correctness relies entirely on test suite outcomes, which may miss semantic issues or partial correctness.
- Evaluation focuses on few agents.
- The system may not support projects requiring non-trivial external setup, and the automatically generated test case may be trivial.

**Questions For Authors:**

- How do you plan to encourage the research community to adopt and test their models on your benchmarks?
- How to integrate SETUPAGENT with existing agent frameworks (e.g., SWE-Agent) to enable seamless evaluation?
- How do you ensure that the generated benchmarks reflect realistic and meaningful developer tasks beyond just passing/failing the generated test cases(maybe trivial)?
- Can SETUPAGENT be extended to support projects with more complex or non-standard setups?

**Relation To Broader Scientific Literature:**

The paper is well-situated within the broader scientific literature on code generation benchmarks and LLM-based code agents.

**Theoretical Claims:**

N.A.

---

> ### Author Rebuttal · Authors · 2025-04-01
>
> We thank the reviewer for their detailed feedback, highlighting the depth of our analysis, the scalability of our approach, and the value of our work for the community. Below, we address their remaining questions and concerns:
>
> **Can you conduct a manual review of generated tasks to assess their quality?**
> We have conducted a manual review of 30 randomly chosen SWA instances for both task and setup quality.
> To assess task quality, we follow the protocol used to create SWE-Verified. That is, we scored issue description quality and clarity on a scale of 0 (well-specified issue with clear success criteria) to 3 (almost impossible to solve correctly without further instructions) and test quality from 0 (test perfectly covers valid solutions) to 3 (tests are too narrow or broad or requiring information not provided in the issue description). For both questions, 0 and 1 are considered passing scores.
> We observe the following:
> 23 (77%) instances have a meaningful and sufficiently complete issue description, and 22 (73%) of these additionally have suitable tests (27 or 90% across all instances) to check whether the issue was fixed.
> To assess setup quality, we score the extract setup as either functionally equivalent to the described setup or incorrect and the test setup from 0 (functionally equivalent to described setup) to 2 (correct tests only partially or not at all executed).
> We observe the following:
> All instances run the correct tests with 77% using exactly the test commands provided in the reference. 22 (73%) instances additionally have a fully correct installation/setup.
> We will include an extended version of these results in a revised version of our paper.
>
> **How do you ensure the generated benchmarks are meaningful beyond passing (trivial) generated tests?**
> We first want to highlight that all tests are taken from the original repositories and were thus written by human contributors and considered to be sufficiently valuable to be merged. We further filter out all instances where not at least one such test that failed before the fix and succeeds after. Finally, our manual evaluation above demonstrates that cases where no meaningful tests are present are rare (only 10%).
>
> **Can you extend the evaluation of the created benchmarks?**
> Yes, please see the response to Reviewer XqFN.
>
> **How can your benchmarks/approach be integrated into existing agent evaluation pipelines?**
> To allow for a seamless integration of our benchmarks, we have made sure to create our benchmark instances such that they are compatible with the popular SWE-Bench evaluation harness (only requiring minor changes to load installation and testing commands from the dataset rather than being hardcoded) and will release both our dataset and the modified version of the harness.
>
> **How do you plan to motivate the community to use your tool and benchmarks?**
> We believe that the research community can benefit strongly from our benchmarks as the increased sample diversity will reduce the effect of overfitting and thus level the playing field for new methods that were not tuned to SWE-Bench at a significant cost. Further, we believe the ability to efficiently create new domain-specific benchmarks will be especially important for practitioners working on domains not well represented in SWE-Bench. We believe these will be strong motivators to adapt the benchmarks/method proposed here. Additionally, we plan to launch a benchmark website similar to swebench.com to allow for easy result tracking and comparison.
>
> **Can you include a direct comparison to ExecutionAgent?**
> We first want to highlight that ExecutionAgent is not capable of setting up specific (historic) states of repositories. It can thus not be used directly for benchmark generation and is limited to our repository setting in Table 3. Directly compared on 25 random repositories considered for SWA-Bench and using GPT4o-mini for both methods, SetupAgent (ours) succeeds on 13, requiring 29.2 s on average (12 min total), while ExecutionAgent succeeds on only 9, requiring on average 2700 seconds (18h total). We thank the reviewer for suggesting this experiment and will include it in the revised version of our paper.
>
> **Can SetupAgent be expanded to handle more complex setups?**
> While SetupAgent can already extract complex setup steps, a key limitation is that our dockerized execution environment does not support running docker instances inside this docker environment. We believe this is an exciting item for future work but out of scope here given the involved engineering challenges. We note that such a version of the benchmark would also be incompatible with existing SWE-Bench evaluation harnesses.
>
> **Conclusion**
> We hope to have addressed the Reviewer's concerns, look forward to their reply, and remain happy to answer follow-up questions.

---

> > ### Comment · Reviewer_FowC · 2025-04-02
> >
> > Thank you for the submission. I would appreciate some clarification on a few points to better understand the contribution.
> >
> > Currently, I couldn't see a clear performance difference between your benchmark and existing SWE datasets. This may be partly due to the selection of baselines—some appear to perform significantly below the state-of-the-art. It would be helpful to include stronger baselines from SWE-bench so that the unique strengths of your benchmark can be more effectively demonstrated.
> >
> > Also, while expanding available resources is valuable, introducing another large, automatically-generated dataset without rigorous analysis could place an additional burden on the community. Ensuring quality and clarity around the dataset’s behavior would make it more useful and trustworthy.
> > Given that the AI community already has SWE-bench, it might be worth considering submitting this work to the Software Engineering (SE) community, where SE benchmark construction and usage are core concerns. Your work could benefit from more targeted feedback and engagement there.

---

> > > ### Author Response · Authors · 2025-04-04
> > >
> > > We thank the reviewer for engaging in the discussion and address their follow-up questions below.
> > >
> > > **Do you observe a performance difference between SWA-, SWEE-, and SWE-Bench?**
> > > Yes! For example, using AutoCodeRover and GPT4o, we observe a 40% lower performance on SWA than on SWE (see Table 6). We have added more experiments in the reply to Reviewer XqFN, reproduced below for convenience. There, we see significant differences for Haiku 3.5 (20%), Llama3.3 (30%), and DeepSeek V3 (70%). We have also added more agents to our evaluation and see that SWE-Agent v1 also exhibits a significant difference (68%) between SWE and SWA.
> > >
> > >
> > > Model comparison with ACR
> > > | |SWA|SWEE|SWE|
> > > |-|-|-|-|
> > > |GPT4o-mini| 8.4% | 9.0% | 8.2% |
> > > |GPT4o| 10.2% | 15.1% | 16.6% |
> > > |Haiku| 10.8% | 12.9% | 13.6% |
> > > |Llama3.3 70B| 8.8% | 10.8% | 12.5% |
> > > |Qwen2.5 72B *| 3% | 2%| 4% |
> > > |DeepSeek V3 *| 8% | 13% | 26% |
> > >
> > > *only evaluated on a random subset of 100 instances
> > >
> > >
> > > Agent comparison with GPT4o-mini
> > > | |SWA|SWEE|SWE|
> > > |-|-|-|-|
> > > |AutoCodeRover v2| 8.4% | 9.0% | 8.2% |
> > > |Openhands| 3.9% | 4.4% | 4.6% |
> > > |SWE-Agent v1|2.6% | 8.9% | 8.2% |
> > > |ZeroShot (Oracle) | 0.9% | 2.2% | 2.8% |
> > > |ZeroShot (BM 25) | 1.3% | 2.8% | 1.5% |
> > >
> > > **Can you add more baselines performing closer to the state-of-the-art?**
> > > Yes, we have added more models and agents, now including the top 3 open source agents on the SWE-Bench-Full leaderboard. We had to evaluate them with cheaper models (GPT4o-mini instead of Sonnet 3.5/3.7, which would be 20x more expensive) due to budget constraints, with current experiments costing already ~5k USD.
> > >
> > > **Is this work a valuable contribution to the AI community beyond SWE-Bench?**
> > > We believe so. SWE-Bench is limited to few repositories and can not be extended automatically, leading to an increasing risk of overfitting and contamination dominating genuine improvements in code agents, potentially misleading the field. For example, DeepSeek V3, in combination with AutoCodeRover, performs much better on SWE than any other evaluated combination but is beaten by multiple other models on SWA-Bench. We have further shown in our manual analysis that SWA is of high quality, supported by some models obtaining very similar performance across datasets. Finally, our benchmark generation framework will allow researchers to create datasets for specific problem domains and thus build and evaluate more specialized software engineering agents.

---

### Official Review · Reviewer_eETZ · 2025-03-22

**Overall Recommendation:** 3

**Summary:**

To achieve the automatic generation of challenging and realistic repository-level coding benchmarks, this work proposes an LLM-driven method, SETUPAGENT, to automate the extraction of valid information from complex real-world repositories, ensuring the correct setup of the environment for perfectly reproducing issues encountered in practice. Using SETUPAGENT, this work constructs two coding benchmarks with different characteristics: SWA-Bench and SWEE-Bench, to evaluate the properties of the generated benchmarks.

**Claims And Evidence:**

I think the main claims have been supported by enough evidence, especially the characters of the generated benchmarks: real-world, diverse, efficient, generalizability.

**Essential References Not Discussed:**

NA

**Experimental Designs Or Analyses:**

In the experiments, the authors tested different code agents and benchmark generators. However, I believe the number of experiments conducted is somewhat limited, especially for benchmark generators. It would be more beneficial to include generators with different capabilities and varying degrees of relatedness, as this would help us better understand the impact of choosing generators with different attributes.

**Methods And Evaluation Criteria:**

Although I am not an expert in this field (LLM-driven coding analysis), I have carefully studied the method proposed by the authors, which involves detailed and well-reasoned sub-processes. Overall, the method appears convincing to me.

**Other Comments Or Suggestions:**

The first line of chapter 5.3 misses a 'W'

**Other Strengths And Weaknesses:**

I think it can be better to further validate the effectiveness of the benchmark, which may include:

1. Whether running SETUPAGENT multiple times on the same set of repositories produces stable benchmarks and whether these benchmarks lead to consistent evaluation results. If not, it implies that the evaluation results of model capabilities may fluctuate due to randomness in the benchmark construction process, making the evaluation unreliable.

2. The automated construction of benchmarks enables good scalability in sample size. Therefore, for this task, it is recommended to determine the optimal number of samples (benchmark size) that balances evaluation efficiency and stability.

**Questions For Authors:**

See above

**Relation To Broader Scientific Literature:**

The automated construction of benchmarks is of great significance to the community.

**Theoretical Claims:**

NA

---

> ### Author Rebuttal · Authors · 2025-04-01
>
> We thank the reviewer for the detailed review and positive feedback, recognizing the relevance and quality of the benchmarks we created. Below, we address their remaining questions:
>
> **Can you assess the repeatability of SetupAgent and what it implies for the generated benchmarks?**
> First, we want to emphasize that benchmarks need only be constructed (and then updated) once to evaluate a large number of code generation methods. Therefore, a lack of repeatability would not have an adverse effect on the benchmark’s value and reliability. Nonetheless, we conducted an additional experiment corresponding to the repository setting in Table 3, rerunning SetupAgent 3 times for 100 random candidate Repositories from SWEE. This led to 26, 27, and 27 successes, where the last two runs install exactly the same instances and the first just drops one. This demonstrates that SetupAgent is highly repeatable. We thank the reviewer for suggesting this experiment and will include it in the revised version of our paper.
>
> **How was the sample size for the generated benchmarks chosen ?**
> We combined our own experience developing code agents and the community's preference for first the Lite and later the Verfied Version of SWE-Bench to choose a sample size of ~500 instances. As the reviewer noted, larger versions can be created easily, however, making evaluation more costly.
>
> **Can you add an experiment analysing the effect of the underlying model on SetupAgent?**
> We first want to highlight that our work focuses on the framework we propose, which we analyse in the ablation presented in Table 4. Further, we specifically chose GPT4o-mini for its great affordability, as it allows us to demonstrate that benchmark generation is now accessible to the wider community. Finally, we note that few cheaper models are available, e.g., even an 8B model hosted by together.ai having the same inference costs, and more capable models being significantly more expensive, e.g., GPT4o having ~15x higher inference costs. We conduct an additional experiment with GPT4o corresponding to 100 SWEE samples in the repository setting in Table 3. We observe GPT4o succeeds for 24 repositories compared to 26 with GPT4o-mini, demonstrating the robustness of our framework to the underlying model. Interestingly GPT4o uses fewer iterations to improve its output (2.1 instead of 3.4 on average), explaining the slightly lower success rate.

---

### Decision · Program_Chairs · 2025-05-01

**Decision:**

Accept (poster)

**Comment:**

**Meta-Review of “Automated Benchmark Generation for Repository-Level Coding Tasks”**

**Summary of Paper**
This paper introduces **SetUpAgent**, a system that automatically configures historically accurate execution environments and extracts test suites for repository-level coding tasks, thereby enabling the creation of large-scale, real-world benchmarks without extensive human intervention. Using SetUpAgent, the authors build two new benchmarks—**SWA-Bench** (focusing on software applications) and **SWEE-Bench** (covering a broad range of less-popular Python repositories)—and compare them against the existing SWE-Bench. They claim that these new benchmarks are both more diverse and more difficult, revealing performance gaps of up to 40% for certain code agents and suggesting that model success rates on SWE-Bench may give a misleading picture of real-world performance.

---

## Strengths

1. **Addresses a Real Bottleneck in Benchmark Creation**
   - Prior repository-level benchmarks rely on heavy manual curation or resort to aggressive filtering that discards complex projects. By contrast, SetUpAgent automates repository installation and test extraction, making it feasible to scale to hundreds of repositories.

2. **Broader and More Diverse Benchmarks**
   - SWE-Bench includes only a small set of popular Python libraries (12 repositories). The newly introduced SWA-Bench (focusing on applications) and SWEE-Bench (focusing on diverse, less-popular projects) include hundreds of repositories and PRs, providing a richer distribution of tasks.
   - Additional experiments (provided in the rebuttal) show model-dependent gaps among SWA, SWEE, and SWE.

3. **Evidence of Distributional Differences**
   - The authors show that issue descriptions, fix complexity, and repository popularity vary notably between the new benchmarks and SWE-Bench. Several code-generation models exhibit substantially lower success on SWA/SWEE (up to 40% using AutoCodeRover on SWA-Bench), highlighting that performance gains on smaller or older benchmarks may not generalize.

4. **Thorough Empirical Analysis**
   - The paper includes:
     - An ablation study of SetUpAgent, measuring which steps (e.g., CI/CD file parsing, iterative improvements) contribute most.
     - A discussion of potential data contamination for some models when popular repositories are used.

---

## Weaknesses and Discussion Points

1. **Concerns About Difference Magnitude and Baselines**
   - Two reviewers felt the performance differences from SWE-Bench were not always large in the main experiments, and that stronger or more numerous baselines could have highlighted the difficulty gap more convincingly. The authors’ rebuttal adds comparisons with more code agents and LLMs.
   - Some reviewers question whether the community needs a benchmark that is not very different from existing one.

2. **Cost and Practicality**
   - Running large-scale repository-level benchmarks can be expensive (authors mention thousands of dollars for just a few model evaluations). At least one reviewer worried that the community might be deterred by computational overheads. While the authors argue cost is inevitable at realistic scale, it remains a real practical concern for adoption.

3. **Comparison to SWE-Bench Verified**
   - SWE-Bench Verified was introduced to mitigate known issues with the original SWE-Bench. Some reviewers wanted more direct discussion of how SetUpAgent’s automatically generated tasks compare in quality to a carefully validated variant such as SWE-Bench Verified.

---

## Recommendation

On balance, authors contributions likely outweigh the reservations. I recommend acceptance.